# GPCR signaling inhibits mTORC1 via PKA phosphorylation of Raptor

Jenna L Jewell[1,2,3]*, Vivian Fu[4,5], Audrey W Hong[4,5], Fa-Xing Yu[6], Delong Meng[1,2,3], Chase H Melick[1,2,3], Huanyu Wang[1,2,3], Wai-Ling Macrina Lam[4,5], Hai-Xin Yuan[4,5], Susan S Taylor[4,7], Kun-Liang Guan[4,5]*

[1]Department of Molecular Biology, University of Texas Southwestern Medical Center, Dallas, United States; [2]Harold C Simmons Comprehensive Cancer Center, University of Texas Southwestern Medical Center, Dallas, United States; [3]Hamon Center for Regenerative Science and Medicine, University of Texas Southwestern Medical Center, Dallas, United States; [4]Department of Pharmacology, University of California, San Diego, La Jolla, United States; [5]Moores Cancer Center, University of California San Diego, La Jolla, United States; [6]Children's Hospital and Institutes of Biomedical Sciences, Fudan University, Shanghai, China; [7]Department of Chemistry and Biochemistry, University of California San Diego, La Jolla, United States

*For correspondence:
Jenna.Jewell@UTSouthwestern.edu (JLJ);
kuguan@ucsd.edu (K-LG)

**Abstract** The mammalian target of rapamycin complex 1 (mTORC1) regulates cell growth, metabolism, and autophagy. Extensive research has focused on pathways that activate mTORC1 like growth factors and amino acids; however, much less is known about signaling cues that directly inhibit mTORC1 activity. Here, we report that G-protein coupled receptors (GPCRs) paired to $G\alpha_s$ proteins increase cyclic adenosine 3'5' monophosphate (cAMP) to activate protein kinase A (PKA) and inhibit mTORC1. Mechanistically, PKA phosphorylates the mTORC1 component Raptor on Ser 791, leading to decreased mTORC1 activity. Consistently, in cells where Raptor Ser 791 is mutated to Ala, mTORC1 activity is partially rescued even after PKA activation. $G\alpha_s$-coupled GPCRs stimulation leads to inhibition of mTORC1 in multiple cell lines and mouse tissues. Our results uncover a signaling pathway that directly inhibits mTORC1, and suggest that GPCRs paired to $G\alpha_s$ proteins may be potential therapeutic targets for human diseases with hyperactivated mTORC1.
DOI: https://doi.org/10.7554/eLife.43038.001

## Introduction

Cells sense their environment and respond by coordinating anabolic and catabolic processes to control cell growth, metabolism, and autophagy. Sufficient nutrients fuel anabolism such as protein synthesis, whereas nutrient deficiency results in catabolism like autophagy. When this sensing mechanism is lost, the end result can be unfavorable, often leading to human disease. The mammalian target of rapamycin (mTOR) is an evolutionarily conserved Ser/Thr kinase that is a key component of a complex referred to as mTOR complex 1 (mTORC1), and is essential in cell growth regulation (*Jewell et al., 2013a*; *Zoncu et al., 2011a*; *Gomes and Blenis, 2015*). Elevated mTORC1 activity is typically seen in many human diseases including cancer, obesity, type two diabetes, neurodegeneration, and metabolic disorders. Besides mTOR, the mTOR Complex 1 comprises regulatory-associated protein of mTOR (Raptor), which aids in mTOR substrate recognition; mammalian lethal with SEC13 protein 8 (mLST8; also known as GβL), which positively regulates mTORC1; and the two negative regulators of mTOR, 40 kDa proline-rich AKT substrate (PRAS40; also referred to as AKT1S1) and DEP domain-containing mTOR-interacting protein (DEPTOR). As the mTOR name infers, a natural product called rapamycin targets and potently inhibits mTORC1. Because high mTORC1 activation is commonly seen in human disease, therapeutics like rapamycin and rapamycin

analogs (rapalogs) that target and inhibit mTORC1, are currently being used in the clinic. However, there are many limitations to rapalogs, for example they are cytostatic rather than cytotoxic and fail to inhibit all mTORC1-mediated processes (*Li et al., 2014*). Thus, by understanding the molecular mechanisms involved in mTORC1 regulation and inhibition, we can further our understanding of cell growth regulation and develop more efficient therapeutics.

Growth factors, amino acids, stress, and energy status are stimuli sensed by mTORC1. Amino acids are thought to be the most potent and are essential for mTORC1 activation. For example, growth factors cannot activate maximal mTORC1 activity when amino acids are limiting (*Sancak et al., 2008*). Through a mechanism not yet entirely clear, elevated amino acid concentrations are thought to promote mTORC1 lysosomal localization, where mTOR comes into close proximity and is activated by the small G-protein Ras homolog enriched in brain (Rheb). Rheb is targeted to the lysosomal surface through the last 15 amino acids, including isoprenylation of its C-terminal CAAX (C = cysteine, A = aliphatic, X = terminal amino acid) box (*Menon et al., 2014*; *Sancak et al., 2010*). Growth factors activate Rheb and mTORC1 by inhibiting the tuberous sclerosis complex (TSC) tumor suppressor. TSC is a GTPase-activating protein (GAP) for Rheb and promotes Rheb-guanosine triphosphate (GTP) hydrolysis, converting Rheb into its inactive guanosine diphosphate (GDP)-bound state (*Inoki et al., 2003*; *Zhang et al., 2003*; *Tee et al., 2003*). Notably, loss-of-function mutations in TSC hyperactivate mTORC1, resulting in TSC and lymphangioleiomyomatosis (LAM) (*Carsillo et al., 2000*; *Inoki and Guan, 2009*; *Kwiatkowski and Manning, 2005*). Patients with TSC and LAM rely on rapamycin as a main pharmacological treatment.

The Rag GTPases were identified to couple amino acid signaling to mTORC1 activation at the lysosome (*Sancak et al., 2008*; *Kim et al., 2008*). In mammals there are four Rag genes: RagA and RagB have high sequence similarity and are functionally redundant, whereas RagC and RagD are highly related in sequence and also redundant. RagA or RagB forms a heterodimer with RagC and RagD, and this dimerization is imperative for mTORC1 activation and Rag protein stability (*Jewell et al., 2015*). RagA or RagB loaded with GTP interacts with the mTORC1 component Raptor at the lysosome. RagA or RagB GTP-bound forms a heterodimer with RagC or RagD loaded with GDP. Interestingly, RagC somatic gain of function missense mutations are found in approximately 17% of patients with follicular lymphoma (*Okosun et al., 2016*).

Cyclic adenosine 3'5' monophosphate (cAMP) is an important second messenger with diverse physiological functions, including cell proliferation and differentiation (*Rocha et al., 2008*; *Dumaz et al., 2002*; *Cho-Chung, 1990*). Upstream of cAMP are seven transmembrane domain receptors called G-protein coupled receptors (GPCRs), which like cAMP have been linked to the development of cancer and have been found highly expressed in multiple cancer cells (*Lappano and Maggiolini, 2011*). The human genome has over eight hundred GPCR members, which are among the most heavily investigated drug targets in pharmaceutical industry, comprising well over 27% of all FDA-approved drugs (*Overington et al., 2006*). Heterotrimeric G-proteins are coupled to GPCRs, and consist of an α-subunit that binds and hydrolyzes GTP as well as β- and a γ-subunits that form a stable complex . GPCRs paired to Gαs proteins are activated in response to their corresponding ligand or hormone and increase cAMP levels. For example, epinephrine or glucagon increases intracellular cAMP levels through interaction with the β2 adrenergic receptor or glucagon receptor, respectively. Ligand-GPCR interaction causes a GPCR conformational change promoting Gαs to be GTP-bound, resulting in dissociation of Gαs from the β- and γ-subunits. The free GTP-bound Gαs then binds to and activates an enzyme called adenylyl cyclase, which catalyzes conversion of ATP to cAMP. Increased cAMP concentrations can lead to activation of cyclic nucleotide-gated channels, guanine exchange proteins activated by cAMP (EPAC), popeye domain containing proteins (Popdc), or protein kinase A (PKA) (*Taylor et al., 2012*; *Taylor et al., 2008*; *Taylor et al., 2004*; *Altarejos and Montminy, 2011*). In the absence of cAMP, PKA is an enzymatically inactive tetrameric holoenzyme consisting of two catalytic subunits and two regulatory subunits (*Hilger et al., 2018*). Downstream of GPCR-Gαs signaling, cAMP activates PKA by cooperatively binding to the regulatory subunits, leading to release and activation of the catalytic subunits. When PKA is activated, it phosphorylates a variety of substrates, preferably on the recognition motif Arg-Arg-X-Ser/Thr-Y (RRXS/TY), where Y tends to be a hydrophobic residue, and Ser/Thr are the phosphorylatable residues (*Taylor et al., 2008*). PKA substrates play diverse roles in multiple signaling cascades and different tissues.

In this study, we report that GPCRs coupled to Gαs proteins can inhibit mTORC1 activity in multiple cell lines and in mice. GPCR-Gαs signaling increases intracellular cAMP levels to activate PKA, and PKA phosphorylates the mTORC1 component Raptor at Ser 791. Functionally, phosphorylation of Raptor at Ser 791 leads to a decrease in mTORC1 signaling and mTORC1-mediated biology.

## Results

### Gαs-coupled GPCRs increase cAMP to inhibit mTORC1 activation and protein synthesis

Despite extensive research mapping signaling cascades that activate mTORC1 like amino acids and growth factors, less is known about signals that can directly inhibit mTORC1. By performing a small GPCR screen, we found that overexpression of GPCRs coupled to Gαs proteins in human embryonic kidney cells (HEK293A) potently inhibited mTORC1 activity (*Figure 1A*). Specifically, expression of the Gαs-coupled GPCRs such as the adrenergic receptor, dopamine receptor D1, and glucagon receptor inhibited mTORC1 activity. In contrast, some GPCRs not coupled to Gαs proteins increased mTORC1 activity, like frizzled homolog D4, lysophosphatidic acid receptor 5, purinergic receptors 1 and 9, and thrombin receptor (*Figure 1A*). Because activation of Gαs-coupled GPCRs increased intracellular cAMP, we treated cells with forskolin, a pharmacological activator of adenylyl cyclase. HEK293A cells were treated with or without forskolin in a dose- and time-dependent manner, and mTORC1 activity was assessed (*Figure 1B–C*). mTORC1 activity was measured by phosphorylation of S6K at Thr 389, a well-characterized mTORC1 substrate. Phosphorylation of cAMP response element-binding protein (CREB) at Ser 133 is a direct target of PKA downstream of GPCR-Gαs-cAMP signaling (*Gonzalez and Montminy, 1989*). As a positive control for elevated cAMP, phosphorylation of CREB at Ser 133 was increased on treating cells with forskolin. Interestingly, mTORC1 activity was potently inhibited after ~30 min in cells stimulated with 1–10 uM forskolin. Short-term forskolin-treated cells did not affect the protein levels of mTORC1 components or other small G-proteins (Rheb, RagA/B/C, Arf1) that regulate the activity of mTORC1 (*Figure 1—figure supplement 1A–B*).

Both biosynthesis and degradation of cAMP regulate intracellular cAMP levels. Adenylyl cyclase elevates cAMP concentrations, whereas phosphodiesterases decrease cAMP levels. Importantly, pharmaceutical drugs are currently used in the clinic as direct phosphodiesterase inhibitors that increase cAMP levels. Treatment of mouse embryonic fibroblast (MEF) cells with the phosphodiesterase inhibitor 3-isobutyl-1-methylxanthine (IBMX), decreased mTORC1 activity similar to that of forskolin (*Figure 1D*). Moreover, forskolin inhibited mTORC1 activity not only in HEK293A and MEF cells, but also in human prostate cancer cells (PC3 and LNcap/AR), human non-small cell lung carcinoma cells (H1299 and H1944), breast cancer cells (MDA-MB-231), haploid cells (HAP1) derived from chronic myelogenous leukemia cells (KBM-7), monkey kidney fibroblasts-like cells (COS7), human cervical cancer derived cells (HeLa), and human pancreatic cancer cells (MIA Paca-2 and PANC-1) (*Figure 1E*).

To exclude the possibility that decreased S6K phosphorylation was the result of activation of a phosphatase when cAMP levels were elevated, we assessed phosphorylation of two other well-known mTORC1 substrates, 4EBP1 phosphorylation at Thr 37/Thr 46 and ULK1 phosphorylation at Ser 758. Similar to S6K, forskolin-treated cells significantly decreased the phosphorylation of both ULK1 and 4EBP1 (*Figure 1F*). mTORC1 positively controls protein synthesis through phosphorylation of S6K and 4EBP1, whereas mTORC1 phosphorylation of ULK1 inhibits autophagy (*Jewell et al., 2013a; Zoncu et al., 2011a*). Because increased cAMP inhibited the phosphorylation of both S6K1 and 4EBP1, we investigated the role of cAMP on protein synthesis (*Figure 1G*). HEK293A cells were treated with or without forskolin, and protein synthesis was measured by $^{35}$S-Met and $^{35}$S-Cys incorporation. Forskolin decreased global protein synthesis by approximately 50%. Taken together, increased intracellular cAMP levels inhibit mTORC1 activity and protein synthesis.

mTOR is a key component not only in mTORC1, but also in mTORC2, which is activated downstream of growth factor signaling (*Figure 1—figure supplement 2A*) (*Jewell and Guan, 2013b; Laplante and Sabatini, 2012*). To investigate whether increased cAMP levels could also inhibit mTORC2, we stimulated HEK293A cells with or without forskolin and analyzed the mTORC2 substrate AKT. Phosphorylation of AKT at Ser 473 was unchanged in HEK293A cells after the addition of forskolin compared to untreated conditions (*Figure 1—figure supplement 2B*)), suggesting that

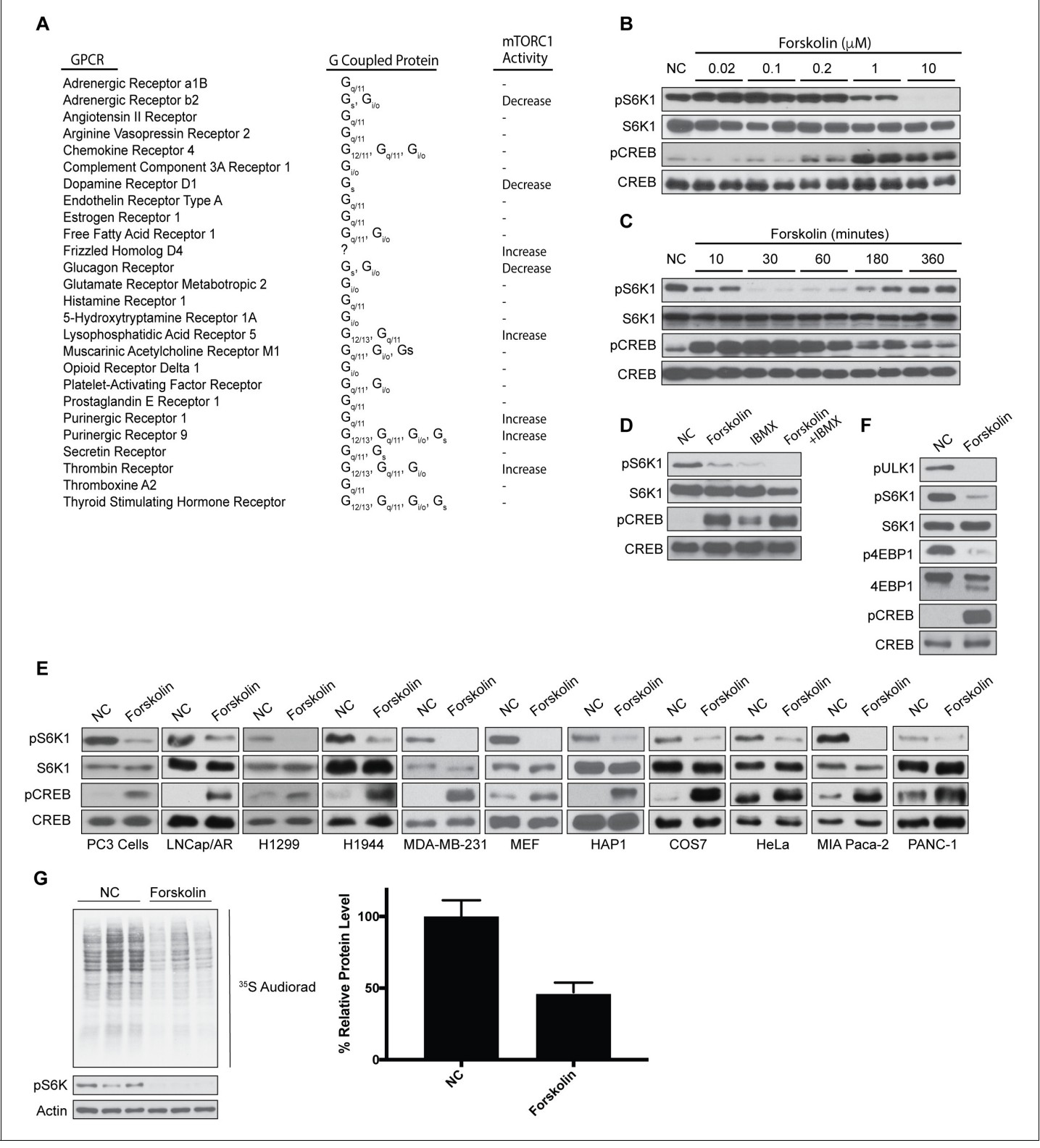

**Figure 1.** Increased levels of cAMP inhibit mTORC1 activation and global protein synthesis. (**A**) Overexpression screen identifying GPCRs involved in mTORC1 regulation. GPCRs were overexpressed in human embryonic kidney 293A (HEK293A) cells, and 24 h later mTORC1 activity was analyzed by protein immunoblotting for the phosphorylation status of S6K1 at Thr 389. mTORC1 activity was visualized to be elevated (Increase), inhibited (Decrease), or unchanged (-). (**B**) Forskolin inhibits mTORC1 in a dose-dependent manner. HEK293A cells were treated with increasing concentrations (µM) of forskolin for 1 h and mTORC1 activity was analyzed by protein immunoblotting for the phosphorylation status of S6K1 (pS6K1) at Thr 389.

*Figure 1 continued on next page*

*Figure 1 continued*

Phosphorylation of CREB (pCREB) at Ser 133 was used as a positive control for the increase of cAMP after forskolin stimulation. Both S6K1 and CREB were used as lysate loading controls. NC denotes normal conditions. (C) Forskolin rapidly and transiently inhibits mTORC1. HEK293A cells were treated with 10 µM forskolin for the indicated time (min) and mTORC1 activity, CREB phosphorylation, and loading controls were analyzed as described in (B). (D) IBMX treatment inhibits mTORC1. Mouse embryonic fibroblasts (MEFs) were treated with 10 µM forskolin or with 10 µM 3-isobutyl-1-methylxanthine (IBMX) alone or in combination for 1 h and mTORC1 activity, CREB phosphorylation, and loading controls were analyzed as described in (B). (E) Forskolin inhibits mTORC1 in multiple cell lines. PC3, LNcap/AR, H1299, H1944, MDA-MB-231, MEF, HAP1, COS7, HeLa, MIA Paca-2, and PANC-1 were treated with or without 10 µM forskolin for 1 h and mTORC1 activity, CREB phosphorylation, and loading controls were analyzed as described in (B). (F) Forskolin inhibits phosphorylation of multiple mTORC1 substrates. HEK293A cells were treated with 10 µM forskolin for 1 h and mTORC1 activity was analyzed by protein immunoblotting for the phosphorylation status of S6K1 (pS6K1) at Thr 389, the phosphorylation status of ULK1 (pULK1) at Ser 758 and the mobility shift of 4EBP1, indicating 4EBP1 phosphorylation. CREB phosphorylation, and loading controls were analyzed as described in (B). (G) Forskolin inhibits global protein translation. HEK293A cells were incubated in Met and Cys-free DMEM with or without 10 uM of forskolin for 50 min. $^{35}$S-labeled L-Met and L-Cys mix was then added to the medium for 10 min and new synthesized proteins were detected by autoradiography (left). Densitometry analysis of each lane as well as actin was performed using ImageJ. Quantification is shown as the decrease of global protein synthesis between normal culture conditions and forskolin treatment. Three independent experiments comparing normal conditions (NC) and forskolin treatment (p=0.0159, t-test, error bars were calculated using SEM).

DOI: https://doi.org/10.7554/eLife.43038.002

The following figure supplements are available for figure 1:

**Figure supplement 1.** Protein levels of the mTORC1 complex or small G-proteins involved in mTORC1 signaling are unaffected by forskolin.

DOI: https://doi.org/10.7554/eLife.43038.003

**Figure supplement 2.** mTORC1 inhibition via increased levels of cAMP is not upstream of the tuberous sclerosis complex.

DOI: https://doi.org/10.7554/eLife.43038.004

cAMP specifically inhibits mTORC1 and not mTORC2. Previous reports have demonstrated that elevated cAMP levels can alter the activity of ERK and AMPK, which are upstream of mTORC1 (*Djouder et al., 2010*; *Zivadinovic and Watson, 2005*). However, we saw no change in the phosphorylation of ERK at Thr 202/Tyr 204 or phosphorylation of AMPK at Thr 172 (*Figure 1—figure supplement 2B–C*). These phosphorylation sites are located within the activation loop of the kinase and are indicative of kinase activity. Also, the phosphorylation of ACC at Ser 79, a well-characterized AMPK substrate, was unchanged in cells treated with forskolin.

## Elevated cAMP activates PKA to block mTORC1 activation

PKA and EPAC mediate most of the physiological functions downstream of cAMP. To investigate whether PKA is involved in cAMP-induced mTORC1 inhibition, we used an ATP-mimetic inhibitor of PKA called N-[2-[[3-(4-Bromophenyl)−2-propenyl]amino]ethyl]−5-isoquinolinesulfonamide (H89). HEK293A, TSC1 +/+, and TSC1-/- cells were pretreated with H89, then stimulated with or without forskolin and mTORC1 activity was analyzed (*Figure 2A–B*). As a positive control, H89 blocked forskolin-induced CREB phosphorylation. TSC comprises TSC1 and TSC2, which form a physical and functional complex downstream of growth factor signaling (*Benvenuto et al., 2000*; *Chong-Kopera et al., 2006*). Pretreatment of cells with H89 prevented mTORC1 inhibition by forskolin. To further confirm the role of PKA in mTORC1 regulation, we overexpressed the HA-tagged PKA catalytic subunit α (HA PKA Catα) and found that mTORC1 activity under normal cell culturing conditions was strongly blocked (*Figure 2C*). Thus, PKA plays a direct role in mTORC1 inhibition.

PKA catalytic subunits form a complex with PKA regulatory subunits when cAMP levels are low, repressing PKA catalytic kinase activity (*Hilger et al., 2018*). In contrast, increased cAMP levels result in cAMP binding to PKA regulatory subunits inducing a conformational change, and releasing the active PKA catalytic subunits. Specific mutations in PKA regulatory subunits Iα and IIα are unresponsive to cAMP levels and cannot release the PKA catalytic subunits, thus they can function as a dominant negative to block PKA activation. EGFP-tagged mutant PKA regulatory subunits Iα and IIα were expressed in HEK293A cells, treated with or without forskolin, and mTORC1 activity was analyzed. Inhibition of PKA by expressing mutant PKA regulatory subunits Iα or IIα in cells, prevented forskolin from decreasing mTORC1 activity (*Figure 2D*). Collectively, our data suggest that PKA mediates the cAMP signal to inhibit mTORC1.

Amino acid and growth factor signaling converge at the lysosome to fine-tune mTORC1 activity. Amino acids promote mTORC1 lysosomal localization where Rheb binds to and activates mTORC1 in response to growth factors (*Jewell et al., 2013a*; *Zoncu et al., 2011a*). To investigate

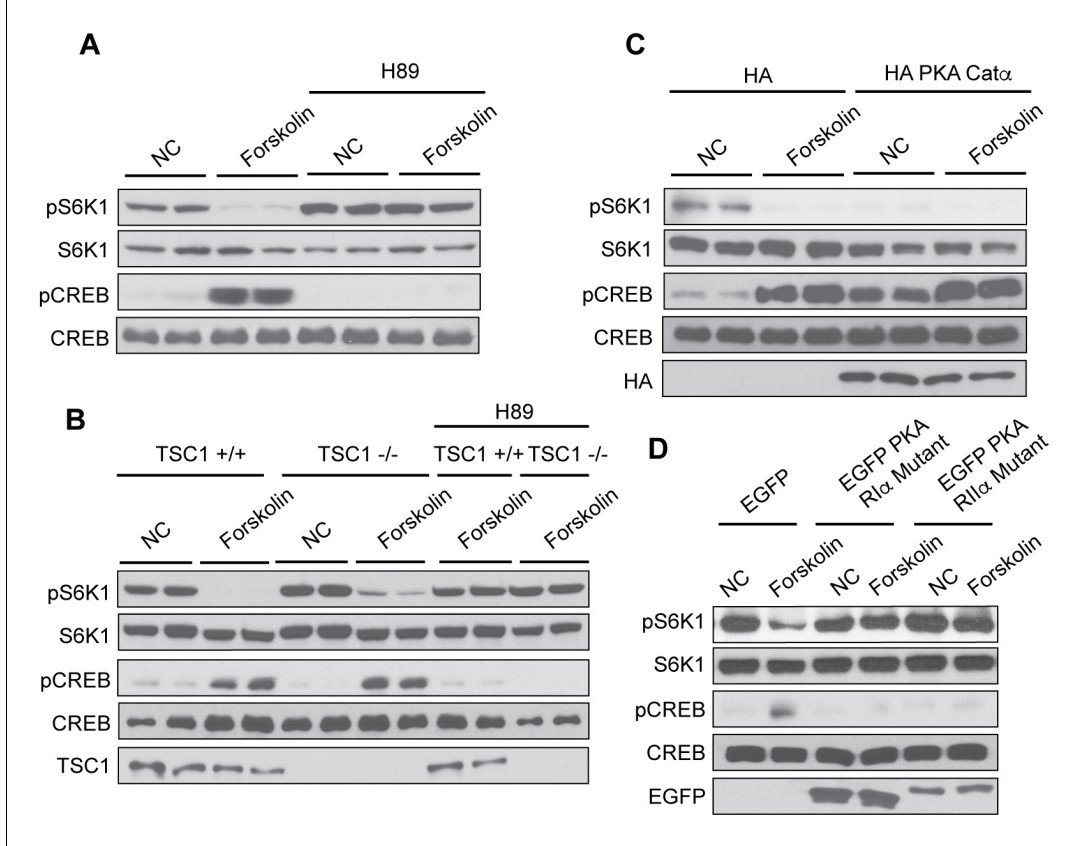

**Figure 2.** Protein Kinase A mediates the effect of cAMP to inhibit mTORC1. (**A**) Inhibition of PKA by H89 blocks the effect of forskolin on mTORC1. Human embryonic kidney 293A (HEK293A) cells were pretreated with or without the protein kinase A (PKA) inhibitor H89, and then treated with or without 10 μM forskolin for 1 h. mTORC1 activity was analyzed by protein immunoblotting for the phosphorylation status of S6K1 (pS6K1) at Thr 389. Phosphorylation of CREB (pCREB) at Ser 133 was used as a positive control for the increase of cAMP after forskolin stimulation. Both S6K1 and CREB were used as lysate loading controls. NC denotes normal conditions. (**B**) PKA is required for forskolin-induced mTORC1 inhibition in TSC1 knockout cells. Control (TSC1 +/+) and tuberous sclerosis complex one knockout (TSC1 -/-) mouse embryonic fibroblasts (MEFs) were pretreated with H89 for 1 h, and then stimulated with or without 10 μM forskolin for 1 h. mTORC1 activity, CREB phosphorylation, and loading controls were analyzed as described in (**A**). TSC1 was also immunoblotted as a control to show the presence or absence of TSC1. NC denotes normal conditions. (**C**) Overexpression of PKA inhibits mTORC1. Empty vector (HA) or HA-tagged PKA Catα was expressed in HEK293A cells. Forty-eight hours later, cells were treated with or without 10 μM forskolin for 1 h. mTORC1 activity, CREB phosphorylation, and loading controls were analyzed as described in (**A**). HA was also immunoblotted as a control to show the presence or absence of HA-tagged PKA Catα. NC denotes normal conditions. (**D**) Dominant negative mutants of PKA regulatory subunits block the effect of forskolin on mTORC1. Empty vector (EGFP) or EGFP-tagged mutant PKA regulatory subunits (PKA RIα or PKA RIIα) were expressed in HEK293A cells. Forty-eight hours later, cells were treated with or without 10 μM forskolin for 1 h. mTORC1 activity, CREB phosphorylation, and loading controls were analyzed as described in (**A**). Expression of EGFP was also immunoblotted as a control to show the presence or absence of EGFP-tagged mutant PKA regulatory subunits (PKA RIα or PKA RIIα) detected by western blotting. NC denotes normal conditions.

DOI: https://doi.org/10.7554/eLife.43038.005

whether increased cAMP levels could inhibit amino acid and growth factor signaling, we pretreated cells with or without forskolin and then stimulated cells with either insulin and/or amino acids (*Figure 3A–B*). Insulin- and amino acid-induced mTORC1 activation was blocked in cells that were pretreated with forskolin. To further dissect whether the cross-talk between cAMP and mTORC1 was through the insulin or amino acid signaling cascade, we performed experiments using TSC1 knock-out (KO) MEF cells (*Figure 3C*). Growth factors act through the TSC complex to activate mTORC1,

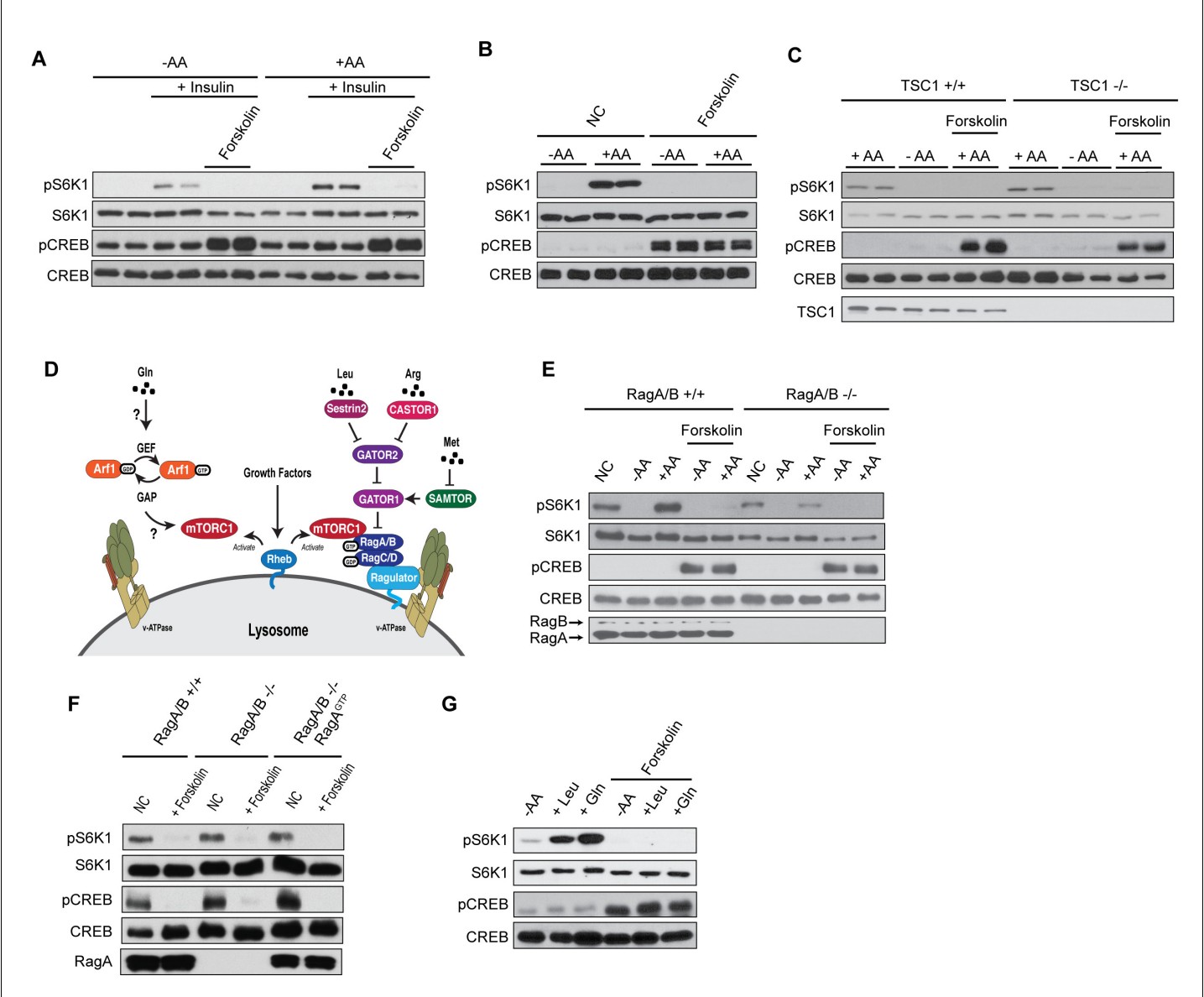

**Figure 3.** cAMP suppresses amino acid-induced mTORC1 activation. (**A**) Forskolin inhibits mTORC1 activation by insulin. Human embryonic kidney 293A (HEK293A) cells were starved of fetal bovine serum (FBS) for 16 h. After FBS starvation cells were starved of amino acids (-AA, left) or not starved of amino acids (+AA, right), pretreated with or without 10 μM forskolin for 1 h, and then stimulated with or without 100 nM insulin for 30 min. mTORC1 activity was analyzed by protein immunoblotting for the phosphorylation status of S6K1 (pS6K1) at Thr 389. Phosphorylation of CREB (pCREB) at Ser 133 was used as a positive control for the increase of cAMP after forskolin stimulation. Both S6K1 and CREB were used as lysate loading controls. (**B**) Forskolin inhibits mTORC1 activation by amino acids. MEFs were starved of amino acids (-AA) for 2 h, pretreated with or without 10 μM forskolin and 10 μM IBMX for 1 h, and then stimulated with or without amino acids (+AA) for 1 h. mTORC1 activity, CREB phosphorylation, and loading controls were analyzed as described in (**A**). NC denotes normal conditions. (**C**) Forskolin inhibits mTORC1 in TSC1 knockout cells. Control (TSC1 +/+) and TSC1 knockout (TSC1 -/-) mouse embryonic fibroblasts (MEFs) were starved of amino acids (-AA) for 2 h, pretreated with or without 10 μM forskolin and 10 μM 3-isobutyl-1-methylxanthine (IBMX) for 1 h, and then stimulated with or without amino acids (+AA) for 1 h. mTORC1 activity, CREB phosphorylation, and loading controls were analyzed as described in (**A**). TSC1 was also immunoblotted to show the presence or absence of TSC1. (**D**) Glutamine activates mTORC1 through a Rag GTPase-independent pathway involving Arf1 (left). Leucine and arginine activate mTORC1 through a Rag GTPase-dependent pathway (right). (**E**) Forskolin inhibits mTORC1 in both wildtype and RagA/B knockout cells. Control (RagA/B +/+) and RagA/B knockout (RagA/B -/-) MEFs were starved of amino acids (-AA) for 2 h, pretreated with or without 10 μM forskolin and IBMX for 1 h, and then stimulated with or without amino acids (+AA) for 1 h. mTORC1 activity, CREB phosphorylation, and loading controls were analyzed as described in (**A**). NC denotes normal conditions. (**F**) Forskolin inhibits mTORC1 in cells expressing RagA, which is constitutively active. Control (RagA/B +/+) MEFs, RagA/B knockout (RagA/B -/-) MEFs, and RagA/B knockout (RagA/B -/- + RagA$^{GTP}$) MEFs expressing a constitutively active RagA were treated with or without 10 μM forskolin and IBMX for 1 h for 1 h. mTORC1 activity, CREB phosphorylation, and loading controls were analyzed as described in (**A**). NC denotes normal

*Figure 3 continued on next page*

*Figure 3 continued*

conditions. (**G**) Forskolin inhibits leucine and glutamine to activate mTORC1. HEK293A cells were starved of amino acids (-AA) for 1 h, pretreated with or without 10 µM forskolin 1 h, and then stimulated with 500 µM Leu (+Leu) or Gln (+Gln) for 1 h. mTORC1 activity, CREB phosphorylation, and loading controls were analyzed as described in (**A**).

DOI: https://doi.org/10.7554/eLife.43038.006

whereas amino acid signaling to mTORC1 is not dependent on TSC (*Jewell et al., 2013a*; *Zoncu et al., 2011a*). Forskolin treatment blocked amino acid-induced mTORC1 activation in both control (TSC1 +/+) and TSC1 KO (TSC1-/-) MEF cells, suggesting that cAMP acts downstream or independent of TSC to inhibit mTORC1.

Different amino acids can activate mTORC1, such as Leu (*Sancak et al., 2008*; *Hara et al., 1998*; *Nicklin et al., 2009*), Arg (*Hara et al., 1998*; *Bauchart-Thevret et al., 2010*), Met (*Gu et al., 2017*), and Gln (*Nicklin et al., 2009*; *Kim et al., 2013*; *Durán et al., 2012*; *van der Vos et al., 2012*). Leu, Arg, and Met require the Rag GTPases (*Gu et al., 2017*; *Wolfson et al., 2016*; *Chantranupong et al., 2016*; *Saxton et al., 2016a*; *Saxton et al., 2016b*; *Chantranupong et al., 2014*), whereas Gln activates mTORC1 independently of the Rag GTPases through Arf1 ((*Jewell et al., 2015*) *Figure 3D*). Both pathways require the v-ATPase, lysosomal function, and mTORC1 lysosomal localization. To determine whether cAMP can inhibit Leu-, Arg-, Met-, or Gln-induced mTORC1 activation, we used RagA/B KO MEF cells (*Jewell et al., 2015*). As mentioned previously, there are four Rag genes in mammals: RagA, RagB, RagC, and RagD. Rag A/B binds to mTORC1, and overexpression of a constitutively active RagA/B (RagA/B bound to GTP) renders mTORC1 insensitive to amino acid starvation conditions. Moreover, deletion of RagA/B significantly diminishes RagC/D protein levels, consistent with RagA/B stabilizing RagC/D by forming hetero-dimers (*Jewell et al., 2015*). Thus, RagA/B KO cells have depleted Rag GTPase complexes. Forsko-lin treatment in both control (RagA/B +/+) and RagA/B KO (RagA/B -/-) MEF cells blocked amino acid-induced mTORC1 activation (*Figure 3E*). Consistently, a constitutively active RagA (RagA$^{GTP}$) reconstituted in RagA/B KO MEFs does not alter cAMP inhibition of mTORC1 (*Figure 3F*). More-over, forskolin treatment in HEK293A cells inhibited both leucine- and glutamine-induced mTORC1 activation (*Figure 3G*). Thus, these results suggest that cAMP inhibits mTORC1, most likely down-stream of Rag GTPases or Arf1.

## PKA activation does not block amino acid-induced mTORC1 lysosomal localization

Increased cAMP levels inhibited both the Leu and Gln signaling pathways to mTORC1 (*Figure 3G*). Because both of these signaling pathways promote mTORC1 lysosomal localization and activation (*Jewell et al., 2015*), we wanted to test whether PKA could block mTORC1 trafficking to the lyso-some. MEF cells were either starved of amino acids (-AA) or stimulated with amino acids (+AA) in the presence or absence of forskolin (*Figure 4A*). As expected, in the absence of amino acids, mTOR did not co-localize to LAMP2-positive lysosomal membranes (*Figure 4A*, first row). In con-trast, the co-localization between mTOR and LAMP2 was seen in response to amino acids (*Figure 4A*, second row). Similarly, in forskolin-treated cells, mTOR also localized to LAMP2-positive lysosomal membranes in response to amino acids. Thus, PKA activation does not block amino acid-induced mTORC1 trafficking to the lysosome.

Amino acid signaling to mTORC1 requires the v-ATPase and lysosomal acidification and function (*Jewell et al., 2015*; *Abu-Remaileh et al., 2017*; *Zoncu et al., 2011b*; *Settembre et al., 2012*). The v-ATPase consists of $V_1$ and $V_0$ domains and is essential for maintaining the low pH necessary for the lysosome to function properly (*Nishi and Forgac, 2002*). Previously it was reported that the v-ATPase $V_{1A}$ subunit was phosphorylated by PKA, and the phosphorylation altered v-ATPase locali-zation (*Alzamora et al., 2010*). To assess whether or not the v-ATPase was involved in mTORC1 inhi-bition by PKA, we investigated lysosomal acidification in the presence or absence of forskolin treatment (*Figure 4B*). MEF cells were treated with or without forskolin and v-ATPase function and lysosomal pH was monitored using Lysotracker, a fluorescent dye used to label acidic organelles. The lysosomal pH was unchanged in forskolin-treated cells when compared to untreated cells. In contrast and as expected, the v-ATPase inhibitor bafilomycin increased lysosomal pH when

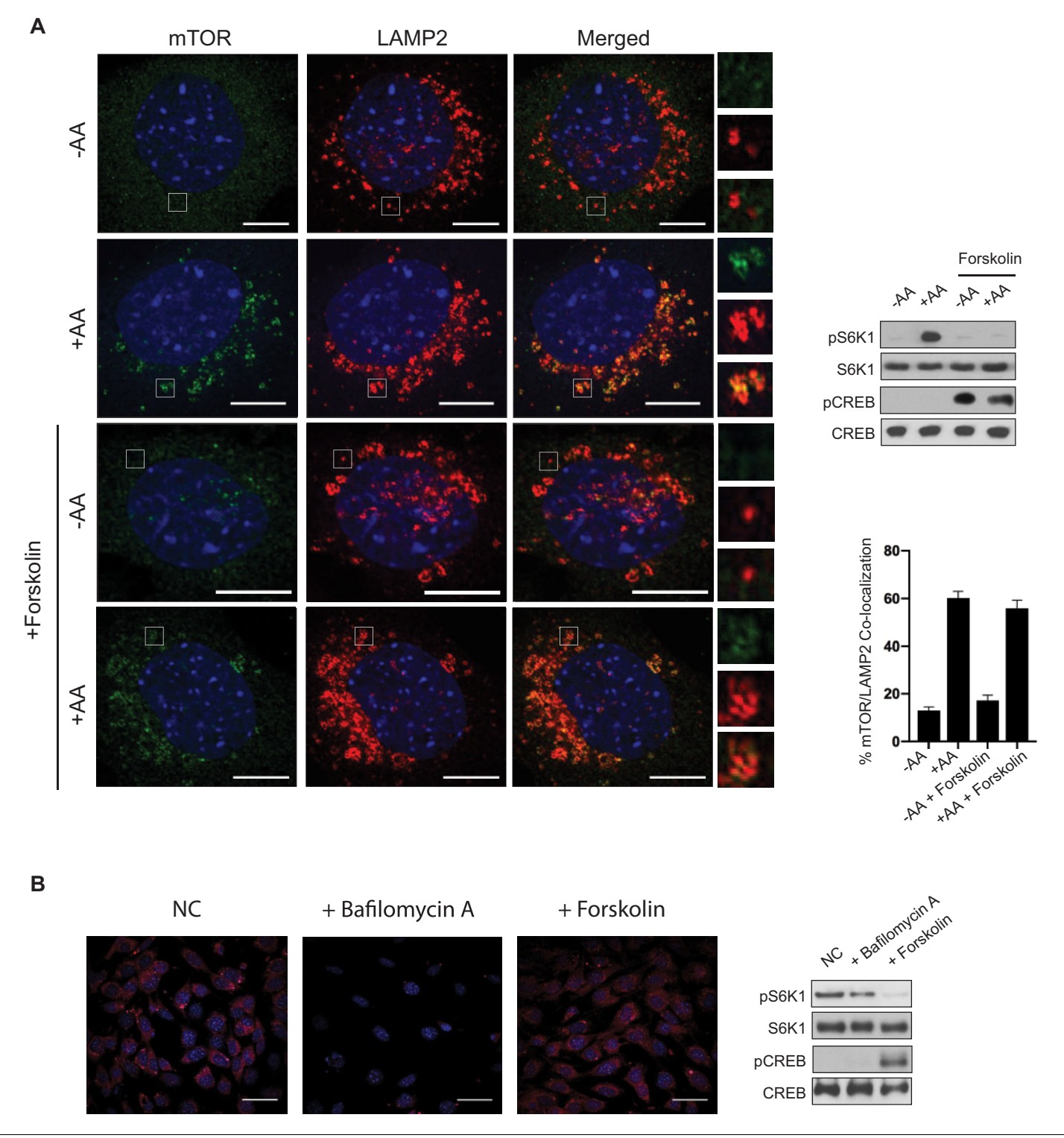

**Figure 4.** Amino acid-induced mTORC1 lysosomal localization is not blocked by cAMP. (**A**) Forskolin does not block amino acid-induced mTOR lysosomal localization. Immunofluorescence analysis depicting mTOR (green) and LAMP2 (red) in mouse embryonic fibroblasts (MEFs). Merged depicts both mTOR and LAMP2. MEFs were starved of amino acids (-AA) for 2 h, treated with or without 10 µM forskolin for 1 h, then stimulated with amino acids (+AA) for 1 h. Higher magnification images of the area depicted by the inset and their overlays are shown on the right. Parallel protein immunoblot as a control for the immunofluorescence images depicting mTORC1 activity via the phosphorylation of S6K1 (pS6K1) is shown (right, top). Phosphorylation of CREB (pCREB) at Ser 133 was used as a positive control for the increase of CREB phosphorylation after forskolin stimulation. Both

*Figure 4 continued on next page*

*Figure 4 continued*

S6K1 and CREB were used as lysate loading controls. The percentages of mTOR/LAMP2 co-localization were quantified for cells pretreated with or without forskolin, and either starved of amino acids (-AA) or stimulated with amino acids (+AA). -AA vs. +AA (p=0.0001, t-test, error bars were calculated using SEM), -AA + forskolin vs.+AA + forskolin (p=0.0001, t-test, error bars were calculated using SEM), -AA vs. -AA + forskolin (p=0.1429, t-test, error bars were calculated using SEM),+AA vs.+AA + forskolin (p=0.3491, t-test, error bars were calculated using SEM) (right, bottom). Scale bar is 10 µm. (B) Forskolin does not appear to alter lysosomal pH. MEFs were treated with or without 10 µM of bafilomycin A or 10 µM forskolin for 1 h, followed by treatment with Lysotracker Red for 15 min. Parallel protein immunoblot a control for the immunofluorescence images depicting mTORC1 activity is shown (right). mTORC1 activity, cAMP levels, and loading controls were analyzed as described in (A). NC denotes normal conditions. Scale bar is 180 µm.

DOI: https://doi.org/10.7554/eLife.43038.007

compared to untreated cells. Taken together, PKA does not inhibit mTORC1 lysosomal localization or lysosome acidification.

## PKA phosphorylates the mTORC1 component Raptor on Ser 791

To investigate whether PKA could directly phosphorylate any components of mTORC1, we used an antibody that recognizes phosphorylated proteins on Ser or Thr residues within the conserved PKA recognition motif R-R-X-S/T-Y (phospho-PKA substrate antibody). Tagged-mTORC1 components (Flag-mTOR, Flag-Raptor, Myc-PRAS40, and Myc-mLST8) were overexpressed in HEK293A cells treated with or without forskolin, immunoprecipitated, and probed with the phospho-PKA substrate antibody (pPKA Sub (RRXS*/T*), *Figure 5A*). Interestingly, in HEK293A cells treated with forskolin, only Raptor was phosphorylated within the conserved PKA recognition motif. Ser 791 on Raptor is the only Ser or Thr that resides within the PKA recognition motif. Raptor Ser 792 was previously reported to be phosphorylated by adenosine monophosphate-activated protein kinase (AMPK), leading to Raptor binding to 14-3-3 and inhibition of mTORC1 (*Gwinn et al., 2008*). Moreover, Raptor Ser 791 is highly conserved across evolution (*Figure 5B*). Architecturally, Ser 791 is located within the WD40 repeat region of Raptor (*Figure 5C*). WD40 repeats typically serve as scaffolds for protein-protein interactions (*Xu and Min, 2011*), suggesting that phosphorylation on Ser 791 within the Raptor WD40 domain may affect the interaction between Raptor and other proteins.

To determine whether Raptor Ser 791 was a PKA phosphorylation site, we performed site-directed mutagenesis on Raptor (*Figure 6A*). HA-tagged Raptor Ser 791 was mutated to either a phospho-defective Ala (Ser791Ala, S791A) or phospho-mimetic Asp (Ser791Asp, S791D). We also mutated HA-tagged Raptor Ser 792 to Ala (Ser792Ala, S792A) and made a double mutant where HA-tagged Raptor Ser 791 and Ser 792 were both mutated to Ala (Ser791Ala/Ser792Ala, S791A/S792A). HA-tagged Raptor or HA-tagged Raptor mutants were overexpressed in HEK293A cells treated with or without forskolin, immunoprecipitated, and probed with the phospho-PKA substrate antibody (pPKA Sub (RRXS*/T*)). Wild-type HA-tagged Raptor was phosphorylated as detected by the pPKA substrate antibody. In contrast, HA-tagged Raptor S791A, HA-tagged Raptor S791D, or HA-tagged Raptor S791A/S792A was not phosphorylated after forskolin treatment. Similar to wild-type HA-tagged Raptor, HA-tagged Raptor S792A was still phosphorylated after forskolin treatment. Forskolin treatment could also inhibit mTORC1 in AMPKα1/2 KO cells (*Figure 6—figure supplement 1A–B*). Consistently, endogenous Raptor immunoprecipitated from HEK293A cells was also phosphorylated as detected by the PKA substrate antibody after the increase of cAMP levels (*Figure 6B*). Reciprocal immunoprecipitation experiments were performed where we used the phospho-PKA substrate antibody to isolate proteins phosphorylated on Ser or Thr residues within the RRXS/TY motif, in the presence or absence of forskolin treatment. The isolated proteins were then immunoblotted and probed for Raptor (*Figure 6C*). Raptor was phosphorylated within the PKA consensus RRXS/TY motif only after forskolin stimulation in wild-type HEK293A cells, but not in PKA Catalytic α/β HEK293A KO cells (PKA Cat α/β KO). PKA Catalytic α/β HEK293A KO cells were generated using CRISPR/Cas9 technology.

We performed *in vitro* kinase assays with recombinant PKA catalytic subunit α to demonstrate that Raptor is a direct substrate of PKA (*Figure 6D*). HA-tagged Raptor, HA-tagged Raptor S791A, or HA-tagged Raptor S792A were immunoprecipitated from HEK293A cells. These proteins were used as substrates for *in vitro* kinase assays with PKA Catα, and Raptor phosphorylation by PKA was determined by immunoblotting with the phospho-PKA substrate antibody. HA-tagged Raptor and

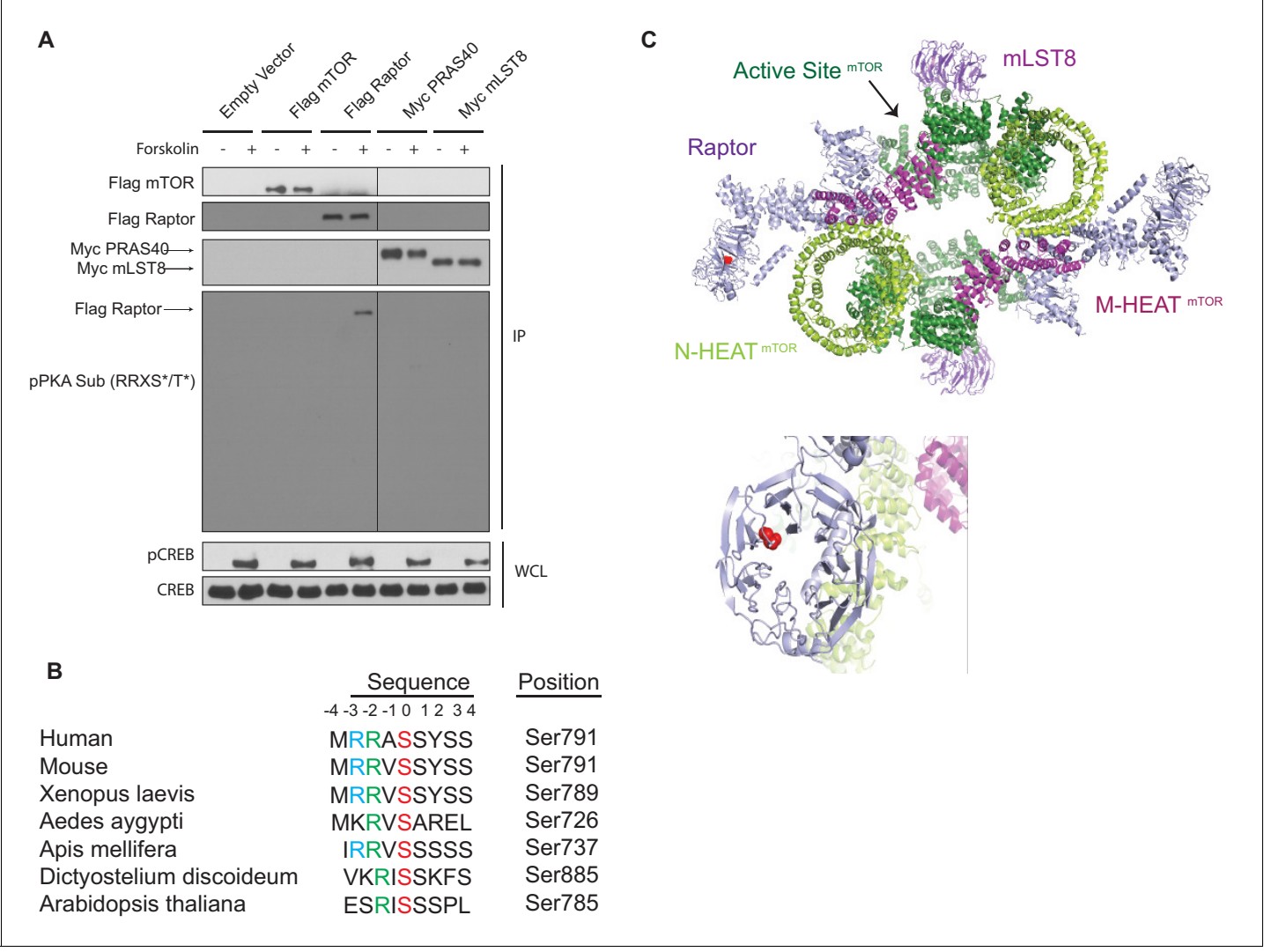

**Figure 5.** cAMP induces Raptor phosphorylation. (**A**) Forskolin increases Raptor phosphorylation detected by a phospho-PKA substrate antibody. Empty vector, Flag-tagged mTOR, Flag-tagged Raptor, Myc-tagged PRAS40, or Myc-tagged mLST8 was expressed in human embryonic kidney 293A (HEK293A) cells. Forty-eight hours later, the cells were treated with or without 10 µM forskolin, and Flag or Myc immunoprecipitates (IPs) were analyzed by immunoblotting with a PKA substrate antibody that recognizes RRXS*/T* (pPKA Sub (RRXS*/T*)). Flag-tagged mTOR and Raptor, or Myc-tagged PRAS40 and mLST8 were also blotted to show equal IPs for cells treated with or without forskolin. CREB phosphorylation was included as a positive control for forskolin stimulation. CREB was used as lysate loading controls. WCL denotes whole cell lysate. (**B**) Alignment of Raptor Ser 791 among different species. (**C**) Raptor Ser 791 resides in the WD40 domain of Raptor. The red ball denotes the phosphorylation.
DOI: https://doi.org/10.7554/eLife.43038.008

HA-tagged Raptor S792A were phosphorylated by PKA, where HA-tagged Raptor S791A was unable to be phosphorylated by PKA. Thus, PKA can directly phosphorylate Raptor on Ser 791.

## Raptor phosphorylation on Ser 791 inhibits mTORC1 signaling

Raptor Ser 791 resides within a WD40 repeat region of Raptor, far away from the kinase domain of mTOR, perhaps suggesting that phosphorylation of Raptor on Ser 791 may alter protein-protein interaction (*Figure 5C*). Consistent with other data (*Figures 3E–G* and *4A*), forskolin treatment or Raptor Ser 791 phosphorylation does not alter the interaction between Raptor and the Rag GTPases (*Figure 6—figure supplement 2*). To test whether Raptor Ser 791 directly altered the kinase activity of mTORC1, we performed mTORC1 in vitro kinase assays (*Figure 6—figure supplement 3A*). HA-tagged Raptor and Myc-tagged mTOR were co-expressed in HEK293A cells treated with or without

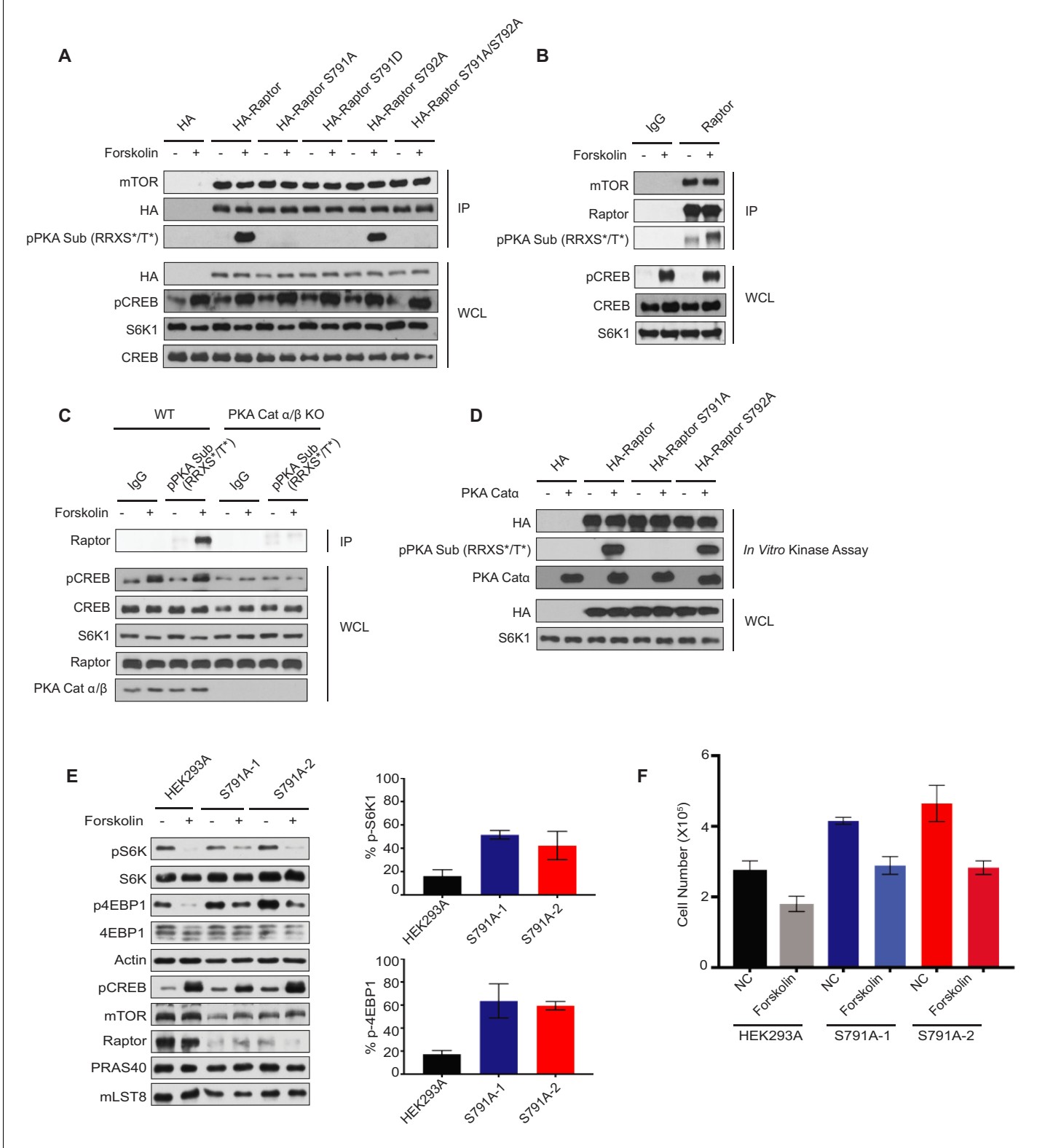

**Figure 6.** Protein kinase A phosphorylates Raptor at Ser 791. (**A**) Forskolin stimulates Ser 791 (S791) phosphorylation in Raptor. HA empty vector, HA-tagged Raptor, HA-tagged Raptor S791A, HA-tagged Raptor S791D, HA-tagged Raptor S792A, or HA-tagged Raptor S791A/S792A was expressed in human embryonic kidney 293A (HEK293A) cells. Forty-eight hours later, the cells were treated with or without 10 μM forskolin, and HA immunoprecipitates (IPs) were analyzed by immunoblotting with a phospho-PKA substrate antibody that recognizes RRXS*/T* (pPKA Sub (RRXS*/T*)).

*Figure 6 continued on next page*

Figure 6 continued

mTOR and HA-tagged Raptor were also blotted to show equal IPs for cells treated with or without forskolin. CREB phosphorylation was included as a positive control for forskolin stimulation. CREB and S6K were used as lysate loading controls. HA was immunoblotted to ensure equal HA expression of the different HA-tagged Raptor constructs. WCL denotes whole cell lysate. (B) Forskolin increases the phosphorylation of endogenous Raptor. HEK293A cells were treated with or without 10 μM forskolin, and Raptor was immunoprecipitated (IP) analyzed by immunoblotting with a PKA substrate antibody that recognizes RRXS*/T* (pPKA Sub (RRXS*/T*)). Raptor and mTOR were also blotted to show equal IPs for cells treated with or without forskolin. Rabbit IgG (IgG) was used as a control. CREB phosphorylation was included as a positive control for forskolin stimulation. CREB and S6K were used as lysate loading controls. WCL denotes whole cell lysate. (C) PKA is required for Raptor phosphorylation by forskolin. HEK293A (WT) or PKA Cat α/β knockout (PKA cat α/β KO) HEK293A cells were treated with or without 10 μM forskolin, and pPKA Sub RRXS*/T* antibody was used for the immunoprecipitation (IP) to enrich for protein kinase A (PKA) substrates. The immunoprecipitations (IP) were analyzed by immunoblotting with Raptor. CREB phosphorylation was included as a positive control for forskolin stimulation. CREB, S6K, and Raptor were used as lysate loading controls. WCL denotes whole cell lysate. PKA Cat α/β was also immunoblotted to confirm the presence or absence of PKA Cat α/β. (D) PKA phosphorylates Raptor S791 directly. HA empty vector, HA-tagged Raptor, HA-tagged Raptor S791A, or HA-tagged Raptor S792A, was expressed in HEK293A cells. Forty-eight hours later, HA immunoprecipitates (IPs) were done and in vitro kinase assays were performed with or without recombinant PKA Cat α. The in vitro kinase assays were analyzed by immunoblotting with a PKA substrate antibody that recognizes RRXS*/T* (pPKA Sub (RRXS*/T*)). HA-tagged Raptor was blotted to show equal IPs. PKA Cat α was also immunoblotted as a control to show the presence or absence of PKA Cat α. HA was immunoblotted to ensure equal HA expression of the different HA-tagged Raptor constructs. WCL denotes whole cell lysate. (E) Right - HEK293A or HEK293A Raptor S791A mutant cells (S791A-1 or S791A-2) were treated with or without forskolin and mTORC1 activity was analyzed by pS6K1 or p4EBP1. S6K and 4EBP1 were loading controls. pCREB was probed for as a positive control indicating the increase in cAMP. The mTORC1 components (mTOR, Raptor, PRAS40, and mLST8) were also immunoblotted for. Left – Quantification of the % decrease of pS6K1 and p4EBP1 after forskolin treatment in HEK293A cells or HEK293A Raptor S791A mutant cells (S791A-1 or S791A-2) from three independent experiments. %pS6K level: HEK293A vs. S791A-1 (p=0.0055, t-test, error bars were calculated by using SEM), HEK293A vs. S791A-2 (p=0.1215, t-test, error bars were calculated by using SEM, increased but not significant). %4EBP1 level: HEK293A vs. S791A-1 (p=0.0382, t-test, error bars were calculated by using SEM), HEK293A vs. S791A-2 (p=0.001, t-test, error bars were calculated by using SEM). (F) Cell number of HEK293A cells or HEK293A Raptor S791A mutant cells (S791A-1 or S791A-2) were quantified 96 h after the initial plating of $5 \times 10^4$ cells per well in normal and forskolin-treated conditions. NC HEK293A vs. S791A-1 (p=0.0071, t-test, error bars were calculated using SEM). NC HEK293A vs. S791A-2 (p=0.0302, t-test, error bars were calculated by using SEM). Forskolin-treated HEK293A vs. S791A-1 (p=0.0308, t-test, error bars were calculated by using SEM). Forskolin-treated HEK293A vs. S791A-2 (p=0.0244, t-test, error bars were calculated using SEM). HEK293A NC vs. forskolin (p=0.0455, t-test, error bars were calculated using SEM). S791A-1 NC vs. forskolin (p=0.0095, t-test, error bars were calculated using SEM). S791A-2 NC vs. forskolin (p=0.0293, t-test, error bars were calculated using SEM).

DOI: https://doi.org/10.7554/eLife.43038.009

The following figure supplements are available for figure 6:

**Figure supplement 1.** Forskolin inhibits mTORC1 in the absence of AMPK.

DOI: https://doi.org/10.7554/eLife.43038.010

**Figure supplement 2.** Raptor Ser 791 phosphorylation does not alter Raptor binding to the Rag GTPases.

DOI: https://doi.org/10.7554/eLife.43038.011

**Figure supplement 3.** cAMP does not alter mTORC1 kinase activity or binding of mTORC1 components.

DOI: https://doi.org/10.7554/eLife.43038.012

**Figure supplement 4.** Generation of the Raptor S791A mutant HEK293A cells using CRISPR/Cas9 genome editing.

DOI: https://doi.org/10.7554/eLife.43038.013

**Figure supplement 5.** Raptor Ser 791 phosphorylation decreases mTORC1 activity and cell proliferation.

DOI: https://doi.org/10.7554/eLife.43038.014

forskolin and mTORC1 was immunoprecipitated (via HA-tagged Raptor) with an HA antibody. *In vitro* kinase assays were performed with immunoprecipitated mTORC1 and the purified mTORC1 substrate 4EBP1. Under normal or forskolin-treated conditions, purified mTORC1 was able to phosphorylate 4EBP1 at Thr 37. However, mTORC1 failed to phosphorylated 4EBP1 when cells were treated with the mTORC1 inhibitor Torin1 (*Thoreen et al., 2009*). Moreover, purified mTORC1 with different Raptor mutants (S791A, S791D, S792D, or S791D/S792D) were also able to phosphorylate 4EBP1 at Thr 37. Collectively, these data suggest that Raptor Ser 791 phosphorylation has no direct effect on mTORC1 kinase activity.

As the phosphorylation of Raptor on Ser 791 does not appear to directly inhibit the kinase activity of mTORC1, we wondered whether Raptor Ser 791 phosphorylation disrupted the binding of components within the mTORC1 complex. To assess changes in mTORC1 protein-protein interactions, we overexpressed HA-tagged Raptor or HA-tagged Raptor phospho-defective and phospho-mimetic mutants (Raptor S791A and Raptor S791D) in HEK293A cells in the presence or absence of forskolin treatment (*Figure 6—figure supplement 3B*). HA-tagged Raptor or HA-tagged Raptor mutants were immunoprecipitated, and the binding of mTORC1 components was analyzed. There

were no significant changes in mTOR, PRAS40, or mLST8 binding to HA-tagged Raptor when compared to the HA-tagged Raptor mutants. Thus, the phosphorylation of Raptor does not appear to impact the Raptor-Rag GTPase binding, the kinase activity of mTORC1 or association of mTORC1 components.

To determine the function of Raptor Ser 791 phosphorylation on mTORC1 signaling, we generated HEK293A cell lines bearing the endogenous Raptor S791A mutation using CRISPR/Cas9 (*Figure 6—figure supplement 4A–B*). The homologous Raptor S791A knock-ins (clones S791A-1 and S791A-2) were confirmed by DNA sequencing. Under normal culture conditions, HEK293A cells with Raptor S791A mutations showed normal mTORC1 signaling in respect to the phosphorylation of S6K and 4EBP1 (*Figure 6E*). The phosphorylation of 4EBP1 was slightly elevated under normal conditions in the Raptor S791A mutant cells when compared to wild-type cells, and the levels of Raptor-S791A and mTOR protein were reduced in the two independent clones when compared to wild-type HEK293A cells. Importantly, forskolin treatment did not inhibit mTORC1 signaling in the Raptor S791A mutant cells to the same degree as in control cells as indicated by the higher phosphorylation of both S6K1 and 4EBP1, two physiological mTORC1 substrates. Treatment of HEK293A cells with forskolin decreased mTORC1 activity ~70% (pS6K1 decreased ~72% and p4EBP1 decreased ~65%) in the control cells. In Raptor S791A mutant cells, forskolin treatment decreased mTORC1 activity anywhere from ~19 to 50% (S791A-1 clone: pS6K1 decreased ~45% and p4EBP1 decreased ~19%; S791A-2 clone: pS6K1 decreased ~50% and p4EBP1 decreased ~31%). Moreover, forskolin decreased pULK1 in HEK293A cells, but not significantly in the Raptor S791A mutant cells (*Figure 6—figure supplement 5A*). These data indicate that phosphorylation of Ser 791 in Raptor is important for effective mTORC1 functional inhibition by forskolin; however, the inhibition of mTORC1 was not completely abolished in the S791A cell lines.

Next, we examined cell proliferation. Cells treated with forskolin or cells expressing PKA Catα had a significant decrease in cell proliferation (*Figure 6—figure supplement 5B–C*). Interestingly, Raptor S791A mutant cells had an increase in cell proliferation when compared to wild-type HEK293A cells under normal and forskolin-treated conditions (*Figure 6F*). Taken together, Raptor S791A mutant cells significantly compromised the ability of cAMP to inhibit mTORC1 function.

## Activation of Gαs-coupled GPCRs inhibits mTORC1 activity

To ensure that our results were physiologically relevant, we turned to different cell lines to activate endogenously expressed Gαs-coupled GPCRs with hormones or agonists. MDA-MB-231 and U20S cells express the Gαs-coupled- GPCRs β1 and β2 adrenergic receptors. We treated MDA-MB-231 and U20S cells with epinephrine, a hormone that binds to and activates β2 adrenergic receptors (*Figure 7A, B*). Similar to forskolin treatment in previous experiments, epinephrine treatment inhibited mTORC1 activation. Also, treatment of MDA-MB-231 cells with the FDA-approved β1 adrenergic receptor agonists, isoprenaline and dobutamine, inhibited mTORC1 activity (*Figure 7A*). Chemical inhibition of PKA with H89 or the knockdown of the PKA Catα subunit with shRNA prevented epinephrine, isoprenaline, and dobutamine from inhibiting mTORC1. Furthermore, mice were injected with or without epinephrine and mTORC1 activity was assessed. mTORC1 activity was significantly decreased in mice brain and liver, a physiological target organ of epinephrine (*Figure 7C, D*), suggesting a physiological role of epinephrine signaling in mTORC1 regulation *in vivo*.

We next examined primary mouse hepatocytes that express the glucagon receptor, another Gαs-coupled GPCR. Isolated primary hepatocytes stimulated with glucagon had significantly decreased mTORC1 activity (*Figure 7E*). Moreover, mTORC1 activity was also decreased in primary hepatocytes stimulated with forskolin, epinephrine, or vasopressin (*Figure 7F*). Vassopressin receptors are Gαs-coupled GPCRs also expressed in the liver. PKA inhibition with H89 prevented the decrease in mTORC1 activity after glucagon, forskolin, epinephrine, or vasopressin stimulation. Taken together, activating Gαs-coupled GPCR signaling cascades inhibit the activation of mTORC1 in vivo.

## Discussion

mTORC1 is often referred to as the 'master regulator' of cell growth because it can sense multiple extracellular and intracellular signals to control cell growth, protein synthesis, autophagy, and metabolism (*Jewell et al., 2013a*; *Zoncu et al., 2011a*; *Gomes and Blenis, 2015*). Here, we report

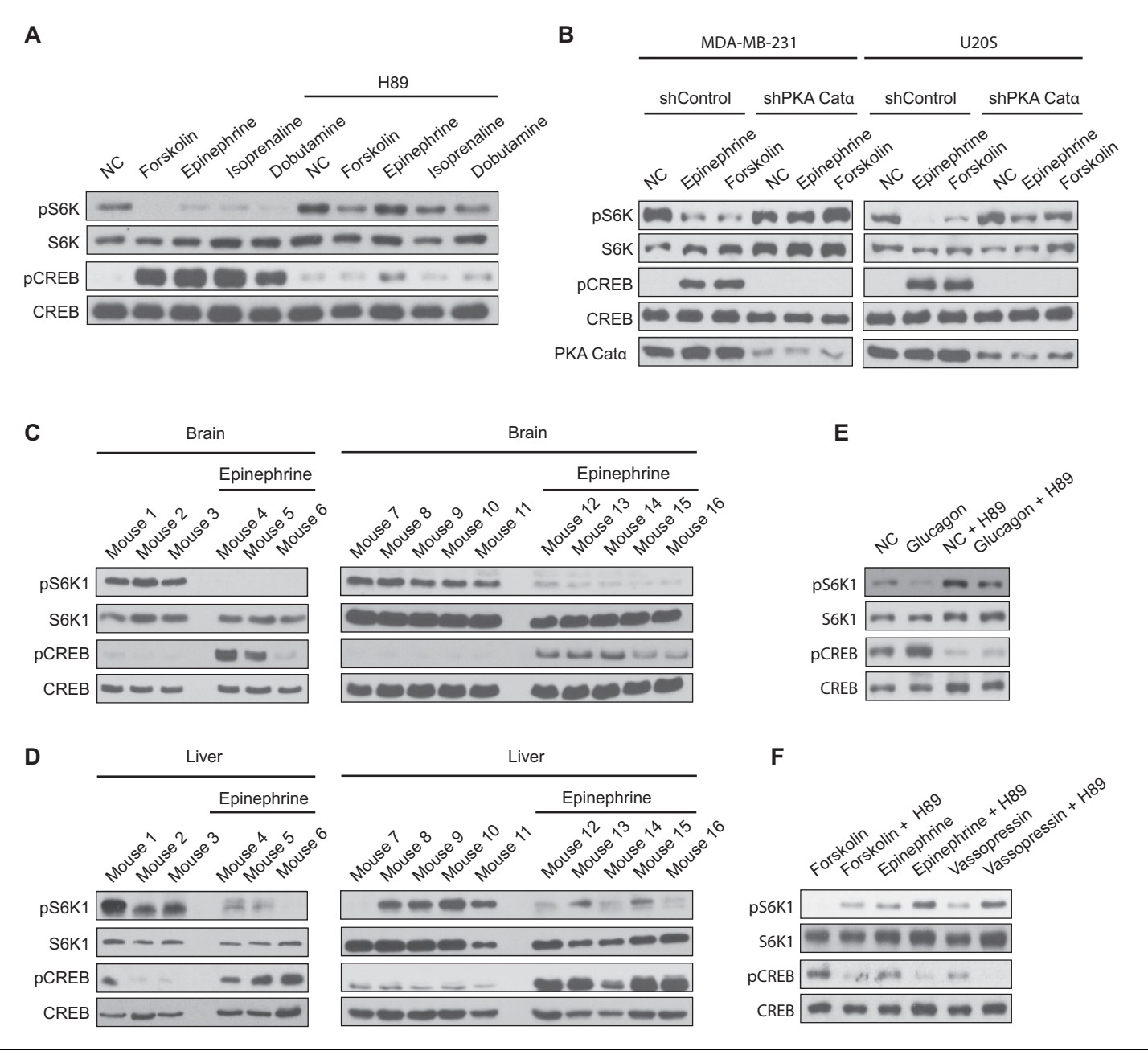

**Figure 7.** Activation of Gαs-coupled GPCRs inhibit mTORC1 activity *in vivo*. (**A**) MDA-MB-231 cells were pretreated with or without the protein kinase A (PKA) inhibitor H89, and then treated with or without 10 µM forskolin, 10 µM epinephrine, 10 µM isoprenaline, or 10 µM dobutamine for 1 h. mTORC1 activity was analyzed by protein immunoblotting for the phosphorylation status of S6K1 (pS6K1) at Thr 389. Phosphorylation of CREB (pCREB) at Ser 133 was used as a positive control for the increase of cAMP after Gαs-coupled GPCR stimulation. Both S6K1 and CREB were used as lysate loading controls. NC denotes normal conditions. (**B**) MDA-MB-231 and U20S stable cell lines expressing control shRNA (shControl) or shRNA targeting the PKA catalytic α subunit (shPKA Cat α) were treated with or without 10 µM forskolin or 10 µM epinephrine. mTORC1 activity and loading controls were analyzed as described in (**A**). PKA Catα was also immunoblotted as a control to show the level of PKA Catα. NC denotes normal conditions. (**C**) Mice were injected with either epinephrine (0.75 ug/g) or propranolol (0.04 mg/g), and 30 min later brain tissues were processed and analyzed for mTORC1 activity by immunoblotting for the phosphorylation status of S6K1 (pS6K1) at Thr 389. Phosphorylation of CREB (pCREB) at Ser 133 was used as a positive control for the increase of cAMP after Gαs-coupled GPCR stimulation (β2 adrenergic receptor). Both S6K1 and CREB were used as lysate loading controls. (**D**) Mice were injected with either epinephrine (0.75 ug/g) or propranolol (0.04 mg/g), and 30 min later livers were processed and mTORC1 activity, cAMP levels (stimulation of β2 adrenergic receptor), and loading controls were analyzed as described in (**A**). (**E**) Primary mouse hepatocytes were treated with 2 µM glucagon for 1 h, and mTORC1 activity, cAMP levels, and loading controls were analyzed as described in (**A**). (**F**)

*Figure 7 continued on next page*

**Figure 7 continued**

Primary mouse hepatocytes were pretreated with or without PKA inhibitor H89, and then treated with 10 μM forskolin, 10 μM epinephrine, or 10 μM vasopressin for 1 h, and mTORC1 activity, cAMP levels, and loading controls were analyzed as described in (A).

DOI: https://doi.org/10.7554/eLife.43038.015

that Gαs-coupled GPCR signaling cascades directly regulate mTORC1 function. GPCRs coupled to Gαs increase cAMP levels and activate PKA, which potently inhibits the activation of mTORC1, protein synthesis, and cell proliferation in multiple cell lines. Our data indicate that PKA directly phosphorylates Raptor on Ser 791. This phosphorylation contributes to functional inhibition of mTORC1 *in vivo*, although the mTORC1 protein kinase activity may not be directly inhibited by Raptor Ser 791 phosphorylation. It is worth noting that the Raptor S791A knock-in HEK293A cell lines did not completely block PKA-induced mTORC1 inhibition. It is possible that PKA phosphorylates another substrate in addition to Raptor at Ser 791 inhibiting mTORC1. Future studies are needed to reveal the precise mechanism of Ser 791 phosphorylation in mTORC1 functional regulation.

Previous studies have reported that increased cAMP levels can regulate mTORC1; however, the exact mechanistic insights are not clear and these studies lack physiological *in vivo* data (*Scott and Lawrence, 1998*; *Xie et al., 2011*; *Monfar et al., 1995*). Moreover, it has been known for some time that increased cAMP levels have anti-proliferative effects in many cancer cell lines (*Rocha et al., 2008*; *Dumaz et al., 2002*; *Chen et al., 2002*; *Naderi et al., 2005*). Epinephrine signaling through the β2 adrenergic receptor, agonists binding to the β1 adrenergic receptor, glucagon signaling through the glucagon receptor, and vasopressin signaling through the vasopressin receptor inhibit mTORC1 activity in cells and *in vivo* (*Figure 7*). Interestingly, glucagon was reported in 1962 as the first hormone to stimulate autophagy in the liver (*Ashford and Porter, 1962*; *Deter and De Duve, 1967*). Later, it was also reported that epinephrine could promote autophagy (*Mortimore and Pösö, 1987*). When mTORC1 activity is high it inhibits autophagy directly through the phosphorylation of ULK1 at Ser 758 (*Kim et al., 2011*). Thus, it is possible that glucagon and epinephrine signaling may promote autophagy through phosphorylation of Raptor at Ser 791 and subsequent mTORC1 inhibition. Because GPCRs are excellent biological targets and are highly druggable, we believe that our results may have important implications for inhibiting mTORC1 function in cancer cells. Furthermore, in addition to agonists that activate Gαs-coupled GPCRs, other FDA-approved drugs alter cAMP levels. For example, many phosphodiesterase inhibitors are currently used in the clinic.

Raptor Ser 791, an evolutionarily conserved residue, resides in the WD40 region of Raptor, a domain well known to facilitate and assemble multi-protein complexes (*Xu and Min, 2011*). Recently, it was reported that PKA phosphorylates Raptor at Ser 791 to promote mTORC1 activation in 3T3L-1 adipocytes and adipose browning in response to β-adrenergic stimulation (*Liu et al., 2016*). We do not have a simple explanation for the discrepancy. Multiple studies likewise show that elevated cAMP levels may lead to activation of mTORC1 (*Blancquaert et al., 2010*; *Kim et al., 2010*). It is worth noting that cAMP is growth inhibitory in many cell lines, but also can be growth stimulatory in other cell lines (*Rocha et al., 2008*; *Dumaz et al., 2002*; *Scott and Lawrence, 1998*; *Xie et al., 2011*; *Monfar et al., 1995*; *Chen et al., 2002*; *Naderi et al., 2005*; *Liu et al., 2016*; *Blancquaert et al., 2010*; *Kim et al., 2010*). Different tissues and cell types may behave differently to cAMP levels in respect to mTORC1 signaling. However, it is worth noting that in ~12 different cell lines we have tested (*Figure 1* and *Figure 7B*), increased cAMP levels always inhibited mTORC1 activity. β-adrenergic signaling in respect to mTORC1 regulation may change in different cell types. Therefore, it appears clear that Gαs-coupled GPCRs activate PKA to phosphorylate Raptor on Ser 791, altering the regulation of mTORC1 and mTORC1-mediated processes.

Raptor has previously been reported to be phosphorylated by other kinases such as AMPK and Nemo-like kinase (NLK). In low energy conditions, AMPK directly phosphorylates Raptor at Ser 792 to induce 14-3-3 binding to Raptor and mTORC1 inhibition. Because Raptor Ser 791 is adjacent to Ser 792, it is possible that Raptor Ser 791 phosphorylation similarly inhibits mTORC1. However, increased PKA activation does not appear to enhance Raptor-14-3-3 binding in HEK293A cells (data not shown). Additionally, Raptor was recently shown to be phosphorylated by NLK at Ser 863 in response to osmotic and oxidative stress signals, leading to a decrease in mTORC1 lysosomal

localization and activation (*Yuan et al., 2015*). It is conceivable to think that mTORC1 regulation by negative signals may converge on Raptor to block the activity of mTORC1 rapidly and efficiently. Therefore, Raptor like TSC, appears to mediate signaling cascades in which mTORC1 is inhibited. We feel that the observations presented here have important implications in nutrient sensing, cell growth control, the mTORC1 field, and cancer biology in general. Because mTORC1 is hyperactivated in many human diseases, targeting Gαs-coupled GPCRs or phosphodiesterases with FDA-approved drugs may be beneficial in inhibiting mTORC1 activity.

## Materials and methods

### Antibodies
The following antibodies were purchased from Cell signaling and used at the indicated dilution for western blot analysis: phospho-S6K1 (#9234, 1:1000), S6K1 (#9202, 1:1000), 4EBP1 (#9452, 1:000), pCREB (#9198, 1:1000), CREB (#9197, 1:1000), pULK1 (#6888, 1:1000), pAKT (#4058, 1:1000), AKT (#9272, 1:1000), pAMPK (#2535, 1:1000), AMPK (#5831, 1:1000), pACC (#11818, 1:1000), ACC (3676, 1:1000#), pERK (#9101), ERK (#9102, 1:1000), mTOR (#2983, 1:1000), Raptor (#2280, 1:1000), PRAS40 (#2610, 1:1000), mLST8 (#3227, 1:1000), Deptor (#11816, 1:1000), RagA (#4357, 1:1000, also recognizes RagB), RagC (#3360, 1:1000), TSC1 (#6935, 1:1000), pPKA Sub (RRXS*/T*) (#9624, 1:1000), and PKA Cat α/β (#4782, 1:1000). EGFP (sc-9996, 1:1000) was obtained from Santa Cruz Biotechnology. Flag (#F3165, 1:5000-1:10000) was obtained from Sigma. Rheb (#H00006009-M01) was obtained from Abnova. Mcherry (#GTX128508, 1:1000) was obtained from Gene Tex. Arf1 (#ARFS 1A9/5, sc-53168, 1:1000) and Myc (#sc-40 HRP were obtained from Santa Cruz). HA (#MMS-101P-500, 1:5000) was from Covance or (sc-7392 or sc-805, 1:5000) Santa Cruz Biotechnology.

Antibodies used for the immunofluorescent microscopy experiments: mTOR (#2983, 1:200) was purchased from Cell signaling. LAMP2 (#13524, 1:200) was obtained from abcam. Secondary antibodies Alexa Fluor 488, 555 used for fluorescent microscopy were obtained from Invitrogen. Lyso-Tracker Red DND-99 (#L7528) was obtained from Life Technologies.

### Chemicals
Forskolin (#1099), 3-isobutyl-1-methylxanthine (IBMX, #2845), and H89 (#2910) were from Tocris. Bafilomycin A1 was from Cayman (#11038). Insulin (#I1507), epinephrine (#E4375), propranolol, (#P0884), isoprenaline (#I6504), dobutamine (#D0676), glucagon (#G2044), and vasopressin (#V0377) were purchased from Sigma. Torin1 was a generous gift from Dr. David Sabatini. Leucine (#L8000) and glutamine (#G3126) were obtained from Sigma.

### Cell lines and tissue culture
Most cell lines (HEK293A (including HEK293A Raptor S791A mutant cells), PC3, H1299, MBA-MD-231, U2OS, MEF (including TSC1-/- and RagA/B -/- MEFs), COS7, HeLa, MIA Paca-2, and PANC-1 cells were maintained at 37°C with 5% $CO_2$, cultured in high-glucose DMEM (#11965–092 from Invitrogen) supplemented with 10% FBS (#F2442 from Sigma), and penicillin/streptomycin (#P0781 from Sigma, 100 units penicillin and 100 μg streptomycin/mL). Most cell lines were from Dr. Kun-Liang Guan's lab (originated from ATCC). These cells were tested using morphology, karyotyping, and PCR-based approaches to confirm their identity - these assays include COI analysis and STR profiling. LNCap/AR cells were from Dr. Ping Mu's lab (originated from Dr. Charles Sawyers' lab) and authenticated by STR profiling. H1944 were from Dr. Katherine O'Donnell's lab (originated from Dr. John Minna's lab) and authenticated by DNA fingerprinting using the PowerPlex 1.2 kit (Promega). COS7 and HeLa cells were from Dr. Vincent Tagliabracci's lab (obtained from ATCC) and authenticated by COI analysis and STR profiling. MIA Paca-2 and PANC-1 cells were from Dr. Rolf Brekken's lab and authenticated by DNA fingerprinting using the PowerPlex 1.2 kit (Promega). We routinely test cells for mycoplasma using Bulldog Bio Inc E-MYCO MYCOPLASMA PCR KIT 48 (#NC0767644 from Fisher).

HAP1 cells were maintained at 37°C with 5% $CO_2$, cultured in IMDM (#I3390 from Sigma) supplemented with 10% FBS (#F2442 from Sigma), and penicillin/streptomycin (#P0781 from Sigma, 100 units penicillin and 100 μg streptomycin/mL).

The generation of the RagA/B KO MEFs has previously been described (*Kim et al., 2014*).

## Primary hepatocyte isolation

All animal experiments were approved by, and conducted in accordance with, an IACUC-approved protocol at UC San Diego (S07213) or UT Southwestern Medical Center (2016–101462).

Primary hepatocytes were isolated from 12 week-old-mice by in situ perfusion with 0.01% collagenase using standard protocol. Crude hepatocytes were purified by Percoll gradient centrifugation. Primary hepatocytes were cultured in complete DMEM medium.

## Epinephrine and glucagon experiments in mice

Mice in experimental and control groups were anesthetized with 0.02 mL/g of Avertin. Once mice were anesthetized, a glucose reading was taken using Accu-Check Nano meter (140–200 mg/dL). Mice were then injected intraperitoneally with epinephrine (0.75 ug/g, dissolved in normal saline), propranolol (0.04 mg/g, dissolved in normal saline), glucagon (1 ug/g, dissolved in 0.05% acetic acid) or saline. Approximately 30 min after the injection, another glucose reading was taken and recorded (~150–250 mg/dL for controls and ~300–500 mg/dL for experimental). Animals were subjected to cervical dislocation, dissected, and tissues were removed and put directly into liquid nitrogen. Frozen tissues were then crushed into powder using tissue pulverizer (Cellcrusher.com). The resulting powdered tissues were weighed, 100 uL/0.01 g lysis buffer (50 mM Tris-HCL, 100 mM NaF, 10 mM EDTA, 2 mM EGTA, 10 mM BetaGP, 1x Protease inhibitor) was added, and then subjected to Polytron (Cat#97057–700 VWR) tissue disruption. Resulting lysates were spun $2 \times 10,500$ rpm for 10 min. Protein concentration was determine using protein assay dye (Bio-Rad #5000006). 20 ul of 5x SDS lysis buffer was added to 80 ul of lysate and then westerns were performed probing for pS6K, S6K, pCREB, and CREB.

## Amino acid starvation and stimulation of cells

Amino acid-free medium was made following the Invitrogen (#11965–092) high-glucose DMEM recipe with the exception that all amino acids were omitted. All experiments with amino acid starvation and stimulation contained 10% dialyzed FBS (#26400–036 from Invitrogen) unless otherwise indicated.

Amino acid starvation (amino acid-free medium and 10% dialyzed FBS) experiments were performed for approximately 1–2 h (HEK293A cells were typically starved for 1 h and MEF cells were starved for 2 h) prior to amino acid stimulation (normal media and 10% dialyzed FBS) for 1 h. For leucine and glutamine stimulation, cells were starved of amino acids, and then stimulated with equal molar (500 μM) leucine and glutamine for 1 h.

For experiments pretreated with forskolin and/or IBMX prior to amino acid stimulation: cells were starved of amino acids for 1–2 h, pretreated with 10 μM forskolin and/or 10 μM IBMX (IBMX was only used in MEF cells) for 1 h, and then stimulated with amino acids for 1 h. For experiments stimulated with insulin, cells were starved of FBS for 16 h followed by the addition of 100 nM insulin for 30 min. For experiments with bafilomycin A, cells were treated with 10 μM bafilomycin A for 1 h.

## Protein synthesis assay

HEK293A cells were washed with PBS and incubated in methionine and cysteine-free DMEM with or without 10 μM forskolin for 1 h. [35]S-labeled L-methionine and L-cysteine mix (75 μCi in a 35 mm dish) (PerkinElmer, NEG772014MC) was then added to the DMEM, and the cells were incubated at 37°C for an additional 10 min. Cells were washed quickly with cold PBS and lysed with sample buffer. Proteins were resolved by SDS-PAGE, and the newly synthesized proteins were detected by autoradiography. Densitometric analysis of each lane as well as actin was performed with Image J. Experiments were repeated three times, and data were shown as the ratio of the densitometric level with forskolin treatment to that without forskolin treatment after normalization to actin.

## Immunofluorescent microscopy

Cells were seeded in 24 well plates on coverslips 1 day prior to experimentation. Coverslips were pretreated with 25 μg/mL of fibronectin (#F1141 from Sigma) at 4°C for 16 h with a quick phosphate-buffered saline (PBS) wash prior to cell seeding. The following steps were performed at room temperature: 4% paraformaldehyde (#2280 from Electron Microscopy Sciences) in PBS was used to fix the cells for 20 min, followed by washing with PBS three times for 5 min each. 0.01% Saponin in

PBS was used to permeabilize the cells for 10 min. Cells on coverslips were blocked in 2% BSA in TBS-Tween for 1 h, followed by washing with TBS-Tween three times for 5 min each. Primary antibodies were diluted in PBS and placed on the cells for 1–3 h, followed by washing with TBS-Tween three times for 5 min each. Secondary antibodies were diluted in PBS and placed on the cells for 1 h, followed by washing with PBS three times for 5 min each, then washed with double distilled water for 5 min. Slides were mounted with prolong gold antifade reagent with DAPI (#P-36931 from Invitrogen). Images were captured with a Zeiss LSM800 confocal microscope. Figures were made with ImageJ and Photoshop.

To determine the co-localization of mTOR and LAMP2 following amino acid starvation and stimulation, the number of mTOR puncta, LAMP2 puncta, and overlapping mTOR/LAMP2 puncta per cell were quantified for each condition. mTOR puncta and LAMP2 puncta were determined by viewing the grayscale image for each channel and outlining punctate structures. mTOR/LAMP2 puncta were determined by overlaying the outlined structures from the mTOR and LAMP2 channels and overlapping structures were denoted as mTOR/LAMP2 puncta. The final percent co-localization value was determined by taking the number of mTOR/LAMP2 puncta and dividing by the total number of LAMP2 puncta per cell. Values represent mean percent co-localization ± SEM. At least five representative cells were analyzed per condition.

For Lysotracker experiments, LysoTracker Red DND-99 (#L7528 from Life Technologies) was added to cells 10 min prior to fixation according to manufacturer's instructions.

## cDNA transfection

Cells were transfected with plasmid DNA using PolyJet DNA In Vitro Transfection Reagent (#50-478-8 from SignaGen Laboratories) according to manufacturer's instructions. For transfection experiments, approximately cells were either plated in 12 well, six well, or 10 cm culture dishes, depending on the particular experiment. Twenty-four hours later, cells were transfected with 500 ng – 2 ug of HA, HA-tagged PKA Catα, EGFP, EGFP-tagged PKA Regulatory subunit mutants (Iα and IIα), Flag, Flag-tagged mTOR, Flag-tagged Raptor, Flag-tagged RagA, Myc-tagged PRAS40, Myc-tagged mLST8, Myc-tagged RagC, HA-tagged Raptor and mutants (S791A, S791D, S792A, and S791A/S792A), Myc-tagged mTOR, mcherry-tagged PKA Regulatory subunit Iα, and Flag-tagged AKAP8. Fresh medium was added 6 h after the transfection, and cells were harvested 24–48 h later.

The HA-tagged Raptor mutants (S791A, S791D, S792A, and S791A/S792A) were made using a QuikChange mutagenesis kit and primers were designed using the following website: http://www.genomics.agilent.com/primerDesignProgram.jsp.

## Cell lysis and immunoprecipitation

Cells were rinsed twice with ice-cold PBS and lysed in ice-cold lysis buffer (40 mM HEPES pH 7.4, 2 mM EDTA, 10 mM pyrophosphate, 10 mM glycerophosphate, and 0.3% CHAPS, and one tablet of EDTA-free protease inhibitors (#11873580001 from Roche) per 25 mL. The soluble fractions from cell lysates were isolated by centrifugation at 13,000 rpm for 10 min in a microfuge. For immunoprecipitations, primary antibodies were added to the lysates and incubated with rotation for 2 h at 4°C. Twenty microliters of a 50% slurry of protein G-sepharose (GE Healthcare #17-0618-01) or A-sepharose (Repligen #IPA-300) were added and the incubation continued for an additional 2 h. In some experiments, just HA-beads (#PI88836, from Fisher) or Myc-beads (#PI20168, from fisher) were used for the immunoprecipitation. Immunoprecipitates were washed three times with lysis buffer. Immunoprecipitated proteins were denatured by addition of 50 µl of sample buffer and boiling for 5 min, resolved by 10–15% SDS-PAGE, and analyzed via western blot analysis.

## Generation of the Raptor S791A mutant knock-in cells

pSpCas9(BB)-2A-Puro (PX459; Addgene plasmid #48139) was a gift from Dr. Feng Zhang (Sanjana et al., 2014). Gene-specific sgRNAs and a homologous recombination (HR) template were designed using Benchling at https://benchling.com. Below are the sgRNA sequences, templates, and primers used for sequencing.

### S791A-1

sgRNA Sequence

5'-ACCGATGAGGGAGTTGAGGG - 3'
Template
5'-AACCTCAGCACCAGCAGCAGCGCCAGCAGCACCCTGGGCAGCCCCGAGAATGAGGAGCA
TATCCTGTCCTTCGAGACCATCGACAAGATGCGCCGCGCCGCCTCCTACTCTTCCCTCAACTCCC
TCATCGGTGAGTCCGCCTGCCCCTTTCTGCTTCCGAGGGGCCCCGAGGGTCTCCTCCCCACACA-
GAGCAGCACAGA - 3'

## S791A-2

sgRNA Sequence
5'-GGGAGGAGTAGGAGCTGGCG - 3'
Template
5'-AACCTCAGCACCAGCAGCAGCGCCAGCAGCACCCTGGGCAGCCCCGAGAATGAGGAGCA
TATCCTGTCCTTCGAGACCATCGACAAGATGAGACGCGCCGCCTCCTACTCCTCCCTCAACTCCC
TCATCGGTGAGTCCGCCTGCCCCTTTCTGCTTCCGAGGGGCCCCGAGGGTCTCCTCCCCACACA-
GAGCAGCACAGA - 3'

## Primers

Forward primer: 5'- GCCAGGGTGTGTGTGTATGA - 3'
Reverse primer: 5'- GAGAACCACCGTTCTGCATC - 3'

HEK293A cells were transfected with sgRNA and HR template, selected with puromycin for 2–3 days, and single-cell sorted by FACs into 96-well plate format. Single clones were expanded and genomic DNA were extracted using Invitrogen 'Purelink Genomic DNA Mini Kit'. Genomic DNA were PCR amplified using the primers indicated above and sent out for sequencing to screen for S791A knock-in clones. PCR primers and sequencing primers were designed 300 bp upstream and downstream of the desired knock-in site. We screened ~70–80 clonal cell lines by sequencing, and only had one homologous (all three alleles repaired the same way) knock-in cell line for each guide (referred to as S791A-1 and S791A-2 in the manuscript).

Densitometric analysis of each pS6K, p4EBP1, and actin was performed with Image J. Experiments were repeated at least three times, and data were shown as the % pS6K1, % p4EBP1, or % pULK1. The % pS6K1, % p4EBP1, or %pULK1 were calculated by comparing normal conditions with forskolin-treated conditions in each of the different cell lines.

## Cell proliferation assays

HEK293A wild-type or Raptor mutant cells (S791A-1 or S791A-2) were seeded at $5 \times 10^4$ cells per well in triplicate in a six-well plate and maintained in media with or without forskolin (10 uM) and IBMX (200 uM). Media was changed every day. At day 4, cells were trypsinized and collected in 1.5 ml tubes and centrifuged for 3 min at 3000 rpm. Cells were resuspended in media then 10 ul was mixed 1:1 with trypan blue. The mixture was then counted using an automated cell counter (BioRad TC20). GraphPad Prism was used to graph results.

For cell proliferation assays overexpressing Flag-tagged PKA Catα (1 μg) and/or Myc-tagged Rheb (1 μg), HEK293A cells were transfected in triplicate in six-well plates, HA-RFP1 were added as a control to bring the total amount to 2 μg for each well. At day 5, cells were trypsinized and collected in 1.5 mL tubes and centrifuged for 3 min at 3000 rpm. Cells were resuspended in media then 10 μL was mixed 1:1 with trypan blue. The mixture was then counted using an automated cell counter (BioRad TC20). GraphPad Prism was used to graph results.

With cell proliferation assays regarding MDA-MB-231 cells, the cells were seeded overnight and treated with or without 10 μM forskolin and 200 μM IBMX or with DMSO as vehicle control (four wells each condition). Media and drugs were replaced every day. At day 5, cells were trypsinized and collected in 1.5 mL tubes and centrifuged for 3 min at 3000 rpm. Cells were resuspended in media then 10 μL was mixed 1:1 with trypan blue. The mixture was then counted using an automated cell counter (BioRad TC20). GraphPad Prism was used to graph results.

## PKA *in vitro* kinase assay

HEK293A cells were transfected with HA-Raptor, HA-Raptor S791A, or HA-Raptor S792A. All Raptor variants were immunoprecipitated as described above, and the immunoprecipitates were incubated

in kinase reaction buffer (50 mM Tris-HCl pH 7.5, 10 mM MgCl$_2$, 10 µM ATP) with or without PKA catalytic subunit (provided by Dr. Susan Taylor, UCSD) at 30°C with shaking for 15 min.

## mTORC1 *in vitro* kinase assay

HEK293A cells were transfected in 10 cm dishes with 4 µg of myc-tagged mTOR and 2 µg of HA-tagged Raptor or HA-tagged Raptor mutants using polyjet transfection reagent. Twenty-four hours later, cells were treated with 10 µM forskolin and 200 µM IBMX, or 1 uM torin as indicated for 50 min. Cells were collected and sonicated in 500 µL of sonication buffer (20 mM Tris, 20 mM NaCl, 20 mM b-glycerol phosphate, 20 mM NaF, 4 mM Na3VO4, 1 mM DTT, protease inhibitor) using a soni-cator (Diagenode Bioruptor) at high intensity and for three cycles of 15 s on and 15 s off. Lysates were spun down at 15,000 rpm for 10 min and supernatant were immunoprecipitated with 10 µl of HA beads (prewashed twice with sonication buffer) for 4 h. Beads were washed once with 25 mM HEPES-KOH pH7.4, 20 mM KCl, 500 mM NaCl, and twice with 25 mM HEPES-KOH pH7.4, 20 mM KCl. Kinase assay was performed at 30°C for 40 min in 50 µL of kinase assay buffer (25 mM HEPES pH7.4, 50 mM KCl, 10 mM MgCl$_2$) with addition of 250 µM ATP (Sigma, #A9187) and 25 ng of recombinant His-tagged 4EBP1 (Fitzgerald Industries International, #80 R-1191) for each reaction.

## RNA interference

Protein expression silencing was done by lentiviral shRNA. Mission shRNA (Sigma Aldrich) was co-transfected with pLKO.1-shRNA-Control or pLKO.1-shRNA-1:(CCGGCAAGTTCCACATTGGTA TCAACTCGAGTTGATACCAATGTGGAATTGGTATCAACTCGAGTTGATACCAATGTGGAACTTG TTTTTTG), together with pSMD.G and psPAX-2 (Addgene) packaging vectors (4.5 µg shRNA, 3 µg, psPAX-2, 1.5 µg psMD.G with 30 µL PolyJet DNA In Vitro Transfection Reagent) into a 10 cm plate to make virus. The medium was changed 6 h later. Forty-eight hours after transfection, the viral supernatant was collected, centrifuged at 3000 x g for 10 min and added to ~50% confluent control and HEK293A cells with 8 µg/mL polybrene (#AL-118 from Sigma Aldrich). Sixteen hours after trans-fection the medium was again changed and puromycin (#ant-pr-1 from Invitrogen) added to a final concentration of 5 µg/mL. Under these conditions, non-infected cells died within 24 h. shRNA used for human PKA Catα (PRKACA) from Sigma:

> TRCN0000001372:CCGGGAAATCCGGGTCTCCATCAATCTCGAGATTGATGGAGACCCGGA
> TTTCTTTTT
> TRCN0000001373:CCGGGATCGAACACACCCTGAATGACTCGAGTCATTCAGGGTGTGTTCGA
> TCTTTTT

## Generation of PKA Catα/β knockout cells using CRISPR/Cas9 genome editing

The 20 nucleotide guide sequences targeting human PRKACA and PRKACB were designed using the CRISPR design tool at http://www.genome-engineering.org/crispr/ (*Hsu et al., 2013*) and cloned into the expression vector SpCas9-2A-Puro V2.0 (pX459) V2.0 (Addgene #62988). The guide sequen-ces targeting Exon 1 of human PRKACA and Exon 9 of human PRKACB are shown below.

> PRKACA
> 3'- AGAACCGCCGCCGCCGCAAC —5'
> PRKACB
> 5'-UAAAAUCGGUCAGUUUCAUC —3'

The single guide RNAs (sgRNAs) in the pX459 vector (500 ng) were transfected into HEK293A cells (six-well) using PolyJet DNA In Vitro Tranfection Reagent according to manufacturer's instruc-tions. Twenty-four hours after transfection, the medium was again changed and puromycin (#ant-pr-1 from Invitrogen) added to a final concentration of 5 µg/mL. Under these conditions, non-infected cells died within 24–48 hr. For the surviving the cells, the medium was changed to medium not con-taining puromycin, and the cells were grown to approximately 80% confluence. The cells were trypsi-nized, washed with PBS, and re-suspended in fluorescence-activated cell sorting (FACs) buffer (PBS, 5 mM EDTA, 2% FBS and Pen/Strep). Cells were single cell sorted by FACs (UCSD; Human Embry-onic Stem Cell Core, BDInflux) into 96-well plate format into DMEM containing 30% FBS and 50 µg/mL penicillin/streptomycin. Single clones were expanded, and screened for PKAα/β by protein

immunoblotting. Refer to the following references for more detail (*Cong et al., 2013*; *Jinek et al., 2012*; *Jinek et al., 2013*; *Mali et al., 2013*).

## Statistical analysis

Statistical analyses were conducted using Prism 7 (Graphpad). Student's t-tests were used for comparison between two groups and p values less than 0.05 were considered as statistically significant.

# Acknowledgements

The authors are grateful to their colleagues in the Guan and Jewell laboratory for critical comments and valuable discussions. We thank Yanhui Xu for incorporating the Ser 791 Raptor phosphorylation site into the mTORC1 complex. We thank Greg Urquhart, Jessica Zhou, Rishika Navlani, and Thu Nguyen for technical help. This work was supported by grants from National Institutes of Health (R01GM51586 and R01CA108941 to KLG, T32CA121938 and R01GM129097-01 to JLJ); The Hartwell Foundation to JLJ; Cancer Prevention Research Institute of Texas (CPRIT) Scholar Recruitment of First-Time, Tenure-Track Faculty Member (RR150032), Cancer Prevention Research Institute of Texas (CPRIT) High-Impact/High-Risk Research Award (RP160713), The Welch Foundation (I-1927–20170325), 2017 UT Southwestern President's Research Council Distinguished Researcher Award, and American Cancer Society Institutional Research Grant (ACS-IRG-17-174-13) to JLJ. CHM is supported by National Institutes of Health grant (T32GM008203) and AWH and VF is supported by National Institutes of Health grant (T32GM007752).

# Additional information

## Competing interests

Kun-Liang Guan: co-founder and has an equity interest in Vivace Therapeutics, Inc, and OncoImmune, Inc. The terms of this arrangement have been reviewed and approved by the University of California, San Diego in accordance with its conflict of interest policies. The other authors declare that no competing interests exist.

## Funding

| Funder | Grant reference number | Author |
| --- | --- | --- |
| Cancer Prevention and Research Institute of Texas | RR150032 | Jenna Jewell |
| Welch Foundation | I-1927–20170325 | Jenna Jewell |
| National Institutes of Health | T32CA121938 | Jenna Jewell |
| National Institutes of Health | R01GM129097 | Jenna Jewell |
| The Hartwell Foundation | | Jenna Jewell |
| Cancer Prevention and Research Institute of Texas | RP160713 | Jenna Jewell |
| University of Texas Southwestern Medical Center | President's Research Council Distinguished Researcher Award | Jenna Jewell |
| American Cancer Society | ACS-IRG-17-174-13 | Jenna Jewell |
| National Institutes of Health | T32GM007752 | Audrey W Hong |
| National Institutes of Health | T32GM008203 | Chase H Melcik |
| National Cancer Institute | | Kun-Liang Guan |
| National Institutes of Health | R01GM51586 | Kun-Liang Guan |
| National Institutes of Health | R01CA108941 | Kun-Liang Guan |

The funders had no role in study design, data collection and interpretation, or the decision to submit the work for publication.

## Author contributions

Jenna L Jewell, Conceptualization, Investigation, Writing—original draft, Writing—review and editing; Vivian Fu, Audrey W Hong, Fa-Xing Yu, Delong Meng, Chase H Melick, Huanyu Wang, Wai-Ling Macrina Lam, Hai-Xin Yuan, Susan S Taylor, Investigation; Kun-Liang Guan, Conceptualization, Investigation, Writing—review and editing

## Author ORCIDs

Jenna L Jewell https://orcid.org/0000-0002-8021-9453
Susan S Taylor https://orcid.org/0000-0002-7702-6108

## Ethics

Animal experimentation: All animal experiments were approved by, and conducted in accordance with, an IACUC approved protocol at UC San Diego (S07213) or UT Southwestern Medical Center (2016-101462).

## Decision letter and Author response

Decision letter https://doi.org/10.7554/eLife.43038.018
Author response https://doi.org/10.7554/eLife.43038.019

## Additional files

### Supplementary files

• Transparent reporting form
DOI: https://doi.org/10.7554/eLife.43038.016

### Data availability

All data generated or analysed during this study are included in the manuscript and supporting files.

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
