## [Decision Letter]

Thank you for submitting your article "Cyclic AMP inhibits mTORC1 via PKA phosphorylation of Raptor" for consideration by *eLife*. Your article has been reviewed by three peer reviewers, one of whom is a member of our Board of Reviewing Editors, and the evaluation has been overseen by Jonathan Cooper as the Senior Editor. The reviewers have opted to remain anonymous.

The reviewers have discussed the reviews with one another and the Reviewing Editor has drafted this decision to help you prepare a revised submission.

Summary:

This study reports that GPCRs paired to Gαs inhibit mTOR via cAMP generation and PKA activation. The authors further show that the mTORC1 component Raptor is phosphorylated by PKA on Ser 791, which may be responsible for mTORC1 inhibition. This work identifies a signaling pathway that inhibits mTORC1 activity and provides potential therapeutic implications for mTORC1-related diseases. There are several points that should be addressed or clarified.

Essential revisions:

1) The link between PKA-mediated Raptor phosphorylation and cAMP-induced mTORC1 inhibition need to be further substantiated. Shown in Figure 6E and F, the K791A knock-in 293 cells seem to have limited effect on preventing the effects of PKA activation on mTOR signaling. This needs to be clarified or further investigated. For example, the exact genotype of these cells should be clearly defined. 293 cells have a complex karyotype (hypotriploid according to ATCC website). Do all copies contain the desired mutation? Or maybe that wildtype alleles remain or small insertions/deletions arise from non-homologous end joining?

How the percentage of pS6K1 or p4EBP1 is calculated should be explained, and the total protein level of 4EBP1 should be included. Additional mTOR downstream targets such as pULK1 or autophagy should be tested. Cell numbers counted in Figure 6F should be examined in the presence and absence of forskolin.

2) As PKA and AMPK phosphorylation occurs on adjacent serine residues in Raptor, a potential coordination of the two signaling pathways should be examined.

Does phosphorylation of Raptor S791 create a 14-3-3 binding site and/or interfere with the binding site generated by AMPK-dependent phosphorylation on S792?

Is pAMPK signal elevated in Figure 1—figure supplement 2? Images with shorter exposure time should be provided. The effect of forskolin treatment on mTORC1 activity should be tested in AMPK knock out cells or in Raptor S792A knock-in cells.

3) How PKA phosphorylation of Raptor inhibits mTORC1 activity warrants further investigation or explanation. What is the implication of impaired mTORC1 signaling without an effect on mTORC1 lysosome recruitment in response to PKA activity? Does Raptor phosphorylation on S791 affect interactions between Raptor and Rags?

Does S791A affect the stability of Raptor? Shown in Figure 6E, the protein levels of mTOR and Raptor reduced greatly in K791A cell lines.

4) Others.mTOR staining on lysosomes should be quantified in Figure 4A. In addition, western blotting in Figure 4A shows that protein level of CREB is significantly elevated in the presence of forskolin, which appears to be contradictory to other data in the manuscript.

A constitutively active RagA/B should be tested in Figure 2 to show cAMP inhibits mTORC1 downstream of RagGTPases.

[Editors' note: further revisions were requested prior to acceptance, as described below.]

Thank you for resubmitting your work entitled "GPCR signaling inhibits mTORC1 via PKA phosphorylation of Raptor" for further consideration at *eLife*.

We appreciate that you have made a substantial effort to address the comments and criticisms of the first review round. We have assessed this revised version with consultation with the two original reviewers.

The manuscript has been improved but there are some remaining issues that need to be addressed before acceptance, as outlined below:

1) The reviewers remained concerned about the effect of Raptor S791A.

a) The explanation of the genotyping of the HEK293 cell lines with the S791A knockin mutation needs to be further clarified as pointed out by reviewer 2 as follows:

The explanation of the genotyping of the HEK293 cell lines with the S791A knockin mutation remains vague. No details are provided concerning how the genotyping was performed. As a result, it is not possible for me to evaluate the claims that the authors were successful in introducing the desired mutation into all copies of the Raptor gene. This has an impact on deciding how to interpret the findings that the S791A mutation only partially protects mTORC1 signaling from PKA-dependent inhibition. The expectation is that these cells have 3 copies of the Raptor gene. How was it determined that all three contain the S791A mutation? If there is heterogeneity in the mutations at this locus, what effect does this have on interpretation of the results.

b) Quantification analyses should be provided in Figure 6—figure supplement 5A and in Figure 6F.

c) The total protein level of 4EBP1 should be provided in Figure 6E, as pointed out by reviewer 3 in the first review round.

d) The issue regarding CREB protein level in Figure 4A should be addressed experimentally.

2) Given that Raptor S791 phosphorylation only partially explains the effect of PKA on mTORC1, this point should be clearly indicated in the Abstract.

3) The result that Forskolin treatment did not lead to increase in Raptor-14-3-3 binding should be acknowledged and discussed (even without showing the data). The current discussion related to this point is in fact contradictory to this result.

4) The statistical analyses should be explained more clearly in both figure legends and in the Materials and methods part, including tests that were used to calculate p-vaules and meanings of the error bars (SD or SEM). Moreover, it would be more informative if individual data points in addition to mean and error bars are shown with the graphs.

---

## [Author Response]

Essential revisions:1) The link between PKA-mediated Raptor phosphorylation and cAMP-induced mTORC1 inhibition need to be further substantiated. Shown in Figure 6E and F, the K791A knock-in 293 cells seem to have limited effect on preventing the effects of PKA activation on mTOR signaling. This needs to be clarified or further investigated. For example, the exact genotype of these cells should be clearly defined. 293 cells have a complex karyotype (hypotriploid according to ATCC website). Do all copies contain the desired mutation? Or maybe that wildtype alleles remain or small insertions/deletions arise from non-homologous end joining?

We have sequenced the RPTOR locus that encodes for S791A, and all alleles (3 alleles) have the desired mutation (Figure 6—figure supplement 4A). We were unable to identify any wild-type alleles or indels suggesting that all copies of the RPTOR gene have been edited to encode the S791A mutation. Because these cell lines (HEK293A S791A) do not completely block PKA-induced mTORC1 inhibition, it is possible that PKA phosphorylates another substrate in addition to Raptor at S791 inhibiting mTORC1. We have inserted a statement about this possibility in the first paragraph of the Discussion.

How the percentage of pS6K1 or p4EBP1 is calculated should be explained, and the total protein level of 4EBP1 should be included.

We calculated the% of pS6K1 in forskolin treated conditions compared to normal culturing conditions in HEK293A cells and in Raptor S791A mutant HEK293A cells (Figure 6E). pS6K1 was normalized to S6K1 protein level. Similar for p4EBP1, where p4EBP1 was normalized to 4EBP1 protein level. We have included these details in the Materials and methods under the Generation of the Raptor S791A mutant knock-in cells section.

The total protein level of 4EBP1 was included in Figure 6—figure supplement 5A.

Additional mTOR downstream targets such as pULK1 or autophagy should be tested.

The phosphorylation of ULK1 (pULK1) was included inFigure 6—figure supplement 5A.Similar to the phosphorylation of S6K and 4EBP1, the pULK1 decreased in wild-type HEK293A cells treated with forskolin (decreased ~50%). In contrast, there was not as much as a decrease in pULK1 in Raptor S791A mutant HEK293A cells when treated with forskolin (S791A-1 = decreased ~7%, S791A-2 = decreased ~0.5%).

Cell numbers counted in Figure 6F should be examined in the presence and absence of forskolin.

Figure 6F has been changed to cell proliferation experiments of the wild-type and Raptor S791A mutant HEK293A cells, in the presence and absence of forskolin.

2) As PKA and AMPK phosphorylation occurs on adjacent serine residues in Raptor, a potential coordination of the two signaling pathways should be examined.Does phosphorylation of Raptor S791 create a 14-3-3 binding site and/or interfere with the binding site generated by AMPK-dependent phosphorylation on S792?

We (both the Guan and Jewell lab) have spent a significant amount of time trying to replicate Raptor binding to 14-3-3 when AMPK is activated (refer to Author response image 1), but cannot. We have tried different conditions that activate AMPK, different tagged 14-3-3 constructs, eluted proteins off immunoprecipitated beads, preblocked the beads with BSA, and precleared cell lysates with beads. Moreover, we don’t see an increase in Raptor-14-3-3 binding in cells treated with Forskolin. From the data in Author response image 1 HA-tagged Raptor binds to GST-tagged 14-3-3 similar to GST (control). In contrast, HA-tagged Yes-associated protein (YAP) binds strongly to GST-tagged 14-3-3 when compared to the control GST. YAP binding to 14-3-3 has previously been shown in the literature (PMID: 12535517, 11118213).

Is pAMPK signal elevated in Figure 1—figure supplement 2? Images with shorter exposure time should be provided.

We included shorter exposures for Figure 1—figure supplement 2C. We don’t see significant changes in pAMPK or the pAMPK substrate Phospho-Acetyl-CoA Carboxylase (pACC).

The effect of forskolin treatment on mTORC1 activity should be tested in AMPK knock out cells or in Raptor S792A knock-in cells.

We treated AMPK α1/2 knockout MEFs and HEK293A cells with Forskolin and assessed mTORC1 activity (Figure 6—figure supplement 1A-B). In both MEFs and HEK293A cells Forskolin inhibited mTORC1 activity.

3) How PKA phosphorylation of Raptor inhibits mTORC1 activity warrants further investigation or explanation. What is the implication of impaired mTORC1 signaling without an effect on mTORC1 lysosome recruitment in response to PKA activity? Does Raptor phosphorylation on S791 affect interactions between Raptor and Rags?

Raptor phosphorylation does not affect the interactions between Raptor and the Rags. We have now included this data (Figure 6—figure supplement 2). Future studies are needed to reveal the precise mechanism of Raptor S791 phosphorylation and mTORC1 signaling.

Does S791A affect the stability of Raptor? Shown in Figure 6E, the protein levels of mTOR and Raptor reduced greatly in K791A cell lines.

We treated HEK293A and Raptor S791A mutant HEK293A (S791A-1 and S791A-2) cells with cycloheximide and MG132 and performed Western blots looking at Raptor protein level (Figure 6—figure supplement 4B, left). Moreover, we performed RT-PCR experiments looking at Raptor mRNA levels (Figure 6—figure supplement 4B, right). S791A-2 had significantly reduced Raptor mRNA levels compared to HEK293A control cells, while S791A-1 did not have significantly reduced Raptor mRNA levels. Blocking translation (cycloheximide) or proteasomal degradation (MG132), didn’t appear to alter Raptor protein levels in HEK293A and Raptor S791A mutant HEK293A cells. Thus, the reason for the reduction in the Raptor S791A mutant HEK293A cells is unclear.

4) Others.mTOR staining on lysosomes should be quantified in Figure 4A. In addition, western blotting in Figure 4A shows that protein level of CREB is significantly elevated in the presence of forskolin, which appears to be contradictory to other data in the manuscript.A constitutively active RagA/B should be tested in Figure 2 to show cAMP inhibits mTORC1 downstream of RagGTPases.

Quantifications (mTOR and LAMP2 co-localization) have been included in Figure 4A.

We (and others) don’t usually see CREB protein level elevated (Figures 1-7 and supplementary figures) with Forskolin treatment. We believe this is just experimental error (perhaps with the Western Blot transfer, etc.).

[Editors' note: further revisions were requested prior to acceptance, as described below.]

The manuscript has been improved but there are some remaining issues that need to be addressed before acceptance, as outlined below:1) The reviewers remained concerned about the effect of Raptor S791A.a) The explanation of the genotyping of the HEK293 cell lines with the S791A knockin mutation needs to be further clarified as pointed out by reviewer 2 as follows:The explanation of the genotyping of the HEK293 cell lines with the S791A knockin mutation remains vague. No details are provided concerning how the genotyping was performed. As a result, it is not possible for me to evaluate the claims that the authors were successful in introducing the desired mutation into all copies of the Raptor gene. This has an impact on deciding how to interpret the findings that the S791A mutation only partially protects mTORC1 signaling from PKA-dependent inhibition. The expectation is that these cells have 3 copies of the Raptor gene. How was it determined that all three contain the S791A mutation? If there is heterogeneity in the mutations at this locus, what effect does this have on interpretation of the results.

We have included the precise details of how we generated the S791A mutant cells and the genotyping below (and in the Materials and methods section), as well as the sequencing results:

“pSpCas9(BB)-2A-Puro (PX459; Addgene plasmid #48139) was a gift from Dr. Feng Zhang (Sanjana et al., 2014). Gene-specific sgRNAs and a homologous recombination (HR) template were designed using Benchling at https://benchling.com. Below are the sgRNA sequences, templates, and primers used for sequencing. […] We screened ~70-80 clonal cell lines by sequencing, and only had 1 homologous (all 3 alleles repaired the same way) knock-in cell line for each guide (referred to as S791A-1 and S791A-2 in the manuscript).”

Our knock-in cells only show a single peak in the sequencing chromograph, suggesting they are homozygous knock-ins (refer to Author response image 2–Author response image 3). We believe that RPTOR has 3 alleles in HEK293A cells because we saw some clonal cell lines repair 3 different ways from some of our other sequencing results (refer to Author response image 4 as an example). Red line indicates the location of expected knock-in site. Blue line indicates the location of PAM sequence.

**Author response image 2. respfig2:** Sequencing chromatogram for knock-in clone S791A-1.

**Author response image 3. respfig3:** Sequencing chromatogram for knock-in clone S791A-2.

**Author response image 4. respfig4:** Sequencing chromatogram for heterozygous knock-in. 3 peaks can be observed in the chromatogram.For example, for the nucleotides underlined in red, you can see 3 distinct peaks for the first 2 nucleotides.

b) Quantification analyses should be provided in Figure 6—figure supplement 5A and in Figure 6F.

Quantifications analyses are now included for pULK1 (Figure 6—figure supplement 5A, left). Quantification analysis for p4EBP1 were already included in Figure 6E.

For Figure 6F, cell number of HEK293A cells or HEK293A Raptor S791A mutant cells (S791A-1 or S791A-2)were quantified 96 hours after the initial plating of 5X10^4^ cells per well in normal and Forskolin treated conditions.

c) The total protein level of 4EBP1 should be provided in Figure 6E, as pointed out by reviewer 3 in the first review round.

The total level of 4EBP1 is now included in Figure 6E.

d) The issue regarding CREB protein level in Figure 4A should be addressed experimentally.

The samples were rerun for Figure 4A, and we included another independent experiment (below). We don’t see significant changes in CREB protein level in Figure 4A, the experiment below, or other similar experiments performed in the manuscript (Figure 3A-3D).

**Author response image 5. respfig5:** 

2) Given that Raptor S791 phosphorylation only partially explains the effect of PKA on mTORC1, this point should be clearly indicated in the Abstract.

We have included this point in the Abstract (see below):

“Mechanistically, PKA phosphorylates the mTORC1 component Raptor on Ser 791, leading to decreased mTORC1 activity. Consistently, in cells where

Raptor Ser 791 is mutated to Ala, mTORC1 activity is partially rescued even after PKA activation.”

3) The result that Forskolin treatment did not lead to increase in Raptor-14-3-3 binding should be acknowledged and discussed (even without showing the data). The current discussion related to this point is in fact contradictory to this result.

We have included this point in the last paragraph of the Discussion:

“Raptor has previously been reported to be phosphorylated by other kinases such as AMPK and Nemo-like kinase (NLK). […] Because mTORC1 is hyperactivated in many human diseases, targeting Gas-coupled GPCRs or phosphodiesterases with FDA approved drugs may be beneficial in inhibiting mTORC1 activity.”

4) The statistical analyses should be explained more clearly in both figure legends and in the Materials and methods part, including tests that were used to calculate p-vaules and meanings of the error bars (SD or SEM). Moreover, it would be more informative if individual data points in addition to mean and error bars are shown with the graphs.

For figures with statistics we included detailed statistical analysis in the figure legend (including tests that were used, p-values, and meanings of error bars). We have also included this information in the Materials and methods section.